# Review article: Retrogressive thaw slump characteristics and terminology

Nina Nesterova[1,2], Marina Leibman[3], Alexander Kizyakov[4], Hugues Lantuit[1,2], Ilya Tarasevich[3,4], Ingmar Nitze[1], Alexandra Veremeeva[1], Guido Grosse[1,2]

[1]Permafrost Research Section, Alfred Wegener Institute Helmholtz Center for Polar and Marine Research, Potsdam, 14473, Germany
[2]Institute of Geosciences, University of Potsdam, Potsdam, 14476, Germany
[3]Earth Cryosphere Institute, Tyumen Scientific Centre SB RAS, Tyumen, postal 625026, Russia
[4]Cryolithology and Glaciology Department, Faculty of Geography, Lomonosov Moscow State University, 119991, Moscow, Russia

*Correspondence to*: Nina Nesterova (nina.nesterova@awi.de)

**Abstract.** Retrogressive thaw slumps (RTSs in plural and RTS in singular) are spectacular landforms that occur due to the thawing of ice-rich permafrost or melting of massive ground ice often in hillslope terrain. RTSs occur in the Arctic, Subarctic as well as high mountain (Qinghai–Tibetan Plateau) permafrost regions and are observed to expand in size and number due to climate warming. As the observation of RTS is receiving more and more attention due to their important role in permafrost thaw, impacts on topography, mobilization of sediment, carbon, nutrients, and contaminants, and their effects on downstream hydrology and water quality, the thematic breadth of studies increases and scientists from different scientific backgrounds and perspectives contribute to new RTS research. At this point, a wide range of terminologies originating from different scientific schools is being used and we identified the need to provide an overview of variable characteristics of RTS to clarify terminologies and ease the understanding of the literature related to RTS processes, dynamics, and feedbacks. We here review the theoretical geomorphological background of RTS formation and landform characteristics to provide an up-to-date understanding of the current views on terminology and underlying processes. The presented overview can be used not only by the international permafrost community but also by scientists working on ecological, hydrological, and biogeochemical consequences of RTS occurrence as well as remote sensing specialists developing automated methods for mapping RTS dynamics. The review will foster a better understanding of the nature and diversity of RTS phenomena and provide a useful base for experts in the field but also ease the introduction to the topic of RTSs for scientists who are new to it.

## 1 Introduction

Permafrost in the Arctic is impacted by thawing in step with ongoing pronounced Arctic warming due to climate change (Biskaborn et al., 2019; Smith et al., 2022). Thaw of ice-rich permafrost results in the formation of characteristic landforms due to sometimes rapid terrain subsidence and erosion. One typical and regionally widespread landform formed by the thaw of ice-rich permafrost or melting of massive ground ice is a slope failure termed retrogressive thaw slump (RTS) (Mackay, 1966). These spectacular landforms in the Northern Hemisphere occur throughout the Arctic, Subarctic, and high mountain

regions (Qinghai–Tibetan Plateau) with ice-rich permafrost and have a significant environmental impact (Kokelj and Lewkowicz, 1999). Figure 1 shows examples of different RTSs photographed across the Northern Hemisphere. RTSs exhibit regional variations in their appearance and characteristics.

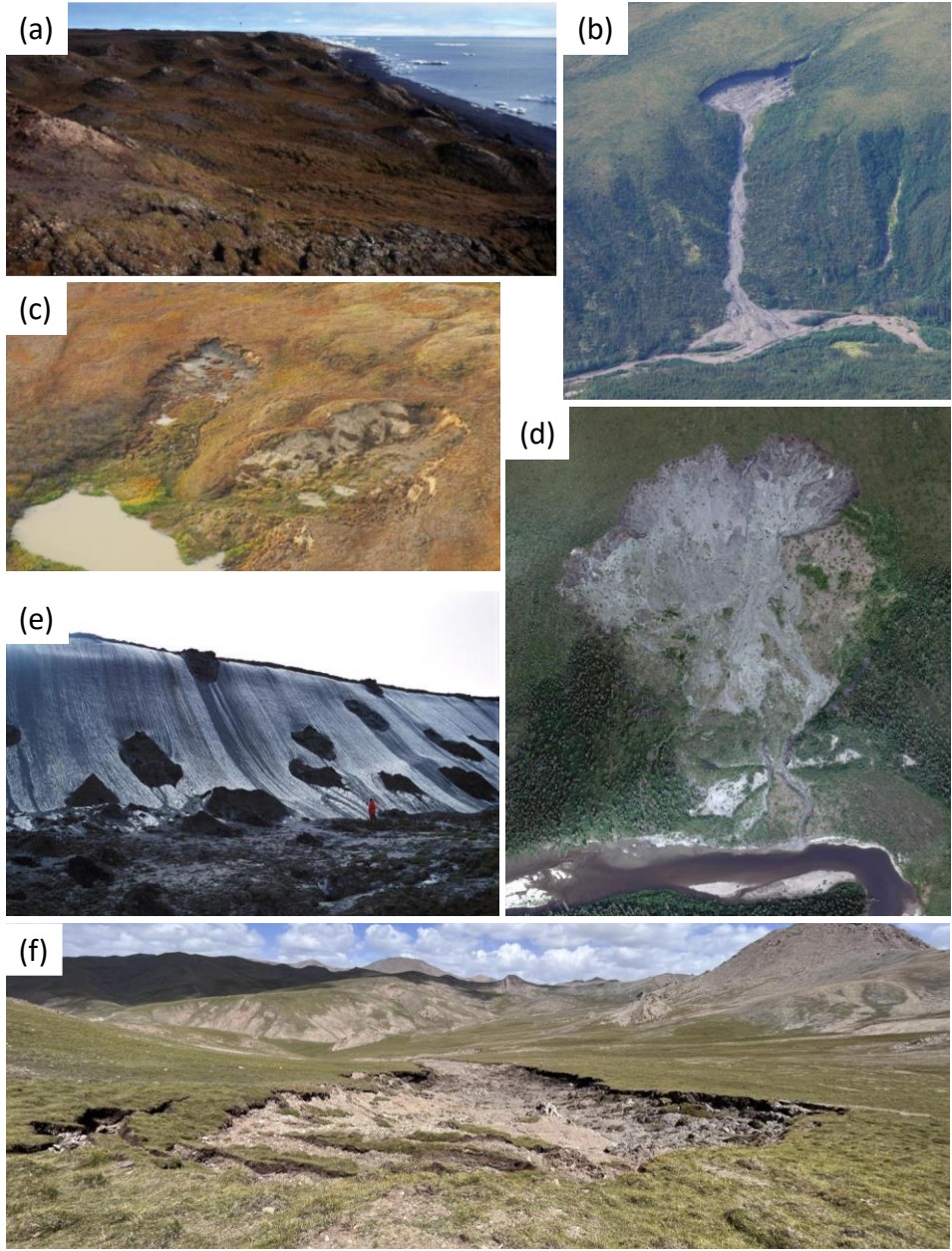

**Figure 1 Various RTSs across the Northern Hemisphere: (a) stabilized yedoma RTS on Belkovsky Island, NE Siberia, Russia, September 2002, photo: Guido Grosse, (b) RTS in the Peel Plateau, NW Canada, July 2023, photo from the airplane: Guido Grosse, (c) two RTSs in the central Gydan Peninsula, West Siberia, Russia, September 2020, photo from helicopter: Elena Babkina, (d) RTS in Selavik, Alaska, USA, July 2021, aerial camera image, credit: AWI, (e) yedoma RTS in Oyagos Yar, NE Siberia, Russia, September 2002, photo: Guido Grosse, (f) RTS in The Qinghai–Tibet Plateau, China, August 2023, photo: Zhuoxuan Xia.**

RTS initiation not only alters the topography, hydrology, and vegetation cover but also contributes to substantial sediment,
carbon, and nutrient fluxes to downstream environments with impacts on water quality and aquatic ecosystems (Kokelj et al.,
2005; Mesquita et al., 2010).
Historically, RTS research started with the first mention of exposed ice in a retrogressive thaw slump probably dating back to
1881 by Dall in his publication on observations in Alaska (Dall, 1881) The first intensive studies on RTSs were conducted
much later in the latter half of the 20th century in Canada (Lamothe and St-Onge, 1961; Mackay, 1966; Kerfoot, 1969) and
Siberia (Popov et al., 1966; Czudek and Demek, 1970). These studies on RTSs were field-based and focused on ground ice,
morphometry, and dynamics. The publications were written either in English or Russian language with different terms applied
to these landforms depending on scientific approaches. Unfortunately, the level of knowledge exchange and reciprocal citation
among RTS researchers from Canada and the USSR was relatively low, leading to the establishment of disparate views and
terminology for RTS used in the literature.
The strong rise in scientific exchange and international collaborations at the end of the 20th century, including joint expeditions
within the permafrost community in general and within the topic of RTS in particular (i.e., Vaikmäe et al., 1993; Ingólfsson,
and Lokrantz, 2003; Are et al., 2005), as well as the emergence of remote sensing methods substantially broadened the scope
of RTS research (Romanenko, 1998; Lantuit and Pollard, 2005; Lantz and Kokelj, 2008; Leibman et al., 2021). Today, a large
body of recent literature predominantly focuses on monitoring RTS activity by measuring retreat rates (Kizyakov et al., 2006;
Wang et al., 2009; Laccelle et al., 2010) and volume changes (Kizyakov et al., 2006; Clark et al., 2021; Jiao et al., 2022;
Bernhard et al., 2022), identifying driving factors (Harris and Lewkowicz, 2000; Lacelle et al., 2010), or more generally
mapping of RTSs (Pollard, 2000; Lipovsky and Huscroft, 2006; Khomutov and Leibman, 2008; Swanson, 2012; Segal et al.,
2016). Recent publications on RTS mapping notably shifted away from a focus on geological and geomorphological aspects
to developing advanced methodologies of RTS detection and classification using spatially and/or temporally high-resolution
remote sensing data and digital elevation data, frequently employing artificial intelligence methods (Huang et al., 2020; Nitze
et al., 2021; Yang et al., 2023).
However, despite the increasing number of studies and strongly rising interest in RTS among the permafrost and remote sensing
research communities, there is still no commonly agreed terminology on the RTS phenomenon. Various authors apply different
terminology to describe the same morphology and processes or use the same terms for different processes. This leads to several
difficulties in communication about RTS within and across research communities. First of all, since the terminology is not
always clearly defined or translated in the literature it can lead to potential misunderstandings about what exact features or
processes have been investigated in a particular study. The confusion about the object of the study may cause incomparability
of the datasets from different RTS studies. Furthermore, different labeling of the same features may result in a completely
different image of the phenomena. For example, Nitze et al. (2024, in review) conducted an experiment where 12 domain
experts from different countries manually mapped RTSs in Canada and Russia. The results demonstrated a large mismatch of
the RTS labeling in Yakutia, Russia, which can be partially explained by different terminology used in the publications
describing this region. The confusion in the terminology and labeling of RTSs can also affect the related studies on how RTSs
impact hydrology, geochemistry, and ecology or their physical modeling, based on the established terms and concepts in the
literature. Moreover, various terms used in the keywords lead to new publications and new data being missed and not included
in further reviews.
This work aims to clarify the existing terminology of RTS phenomena and ease the understanding of published studies. The
paper presents commonly observed RTS characteristics and a neutral review of existing RTS terminology in the literature. Our
review considers a broad variety of RTSs in the Northern Hemisphere.

## 2 Observed characteristics of retrogressive thaw slumps

### 2.1. Morphometry and dynamics

RTSs can be of various sizes starting from less than 0.1 ha in area and reaching up to ~80 ha as the Batagay slump, Yakutia,
Russia (Kizyakov et al., 2023). Some authors describe RTSs larger than 5 ha (Kokelj et al., 2015) or larger than 20 ha (Lacelle
et al., 2015) as megaslumps. Known RTS headwall (in the meaning of the entire steep wall) heights range from a few meters
up to 55 m in Batagay slump (Kizyakov et al., 2023). RTS headwall length can exceed 1 km as reported for Yakutia, Russia
(Costard et al., 2021).
Reported length-to-width ratios range from below 1 (Lantuit and Pollard, 2008; Ardelean et al., 2020) up to 3 (Niu et al., 2016)
and even 5 (Lantuit and Pollard, 2008). Some field studies in Canada suggest that this ratio increases with time due to the
headwall retreating faster than the sidewalls, leading to a landform lengthening (Lewkowicz, 1987b). Other studies in Siberia
report the widening of RTSs with time due to their merging with neighboring RTSs (Runge et al., 2022; Leibman et al., 2023).
RTS dynamics can be estimated by measuring headwall retreat rates. Reported RTS headwall retreat rates vary from several
cm per year in Qinghai–Tibetan Plateau, China (Sun et al., 2017) to up to ~66 m per year estimated for Yugorsky Peninsula,
Russia (Leibman et al., 2021). Similar extreme headwall retreat rates of ~27 m per year were reported for some RTS in Canada
(Lacelle et al., 2015; Jones et al., 2019).

### 2.2. Position and topography

RTSs form inland or on the coasts. Inland RTS can be found at lake shores, riverbanks, slopes of temporary streams, or valley
slopes. As an RTS develops, the thawed material is transferred downstream to valley bottoms or the nearest water body.
The extent of RTSs appearing at a particular position varies and is strongly controlled by the topographical and geological
characteristics of the area. For example, RTSs in mountain regions mostly occur on slopes unrelated to the waterbodies, as in
the Qinghai-Tibetan Plateau, China (Niu et al., 2012; Huang et al., 2018; Hu et al., 2019), or in the Yana Highlands, East
Siberia (Kizyakov et al. 2023), while RTSs in the flat terrain of the Yamal and Gydan peninsulas, West Siberia, are generally
found next to lake shores (Nesterova et al., 2021). A first analysis across the Arctic has not revealed any correlations between
the influence of RTS position in the terrain and its size or activity so far (Bernhard et al., 2022).

RTSs were found across a wide range of slopes, including on gentle terrain slopes of <5° (De Krom, 1990; Leibman et al., 2023), medium slopes of 5 to 10° (Niu et al., 2016), as well as on steep slopes >10° (Czudek and Demek, 1970; Barry, 1992; Robinson, 2000). Some studies found that RTSs on steeper slopes tend to have higher headwall retreat rates (see Sect. 3.1.1) than those that occur on less steep slopes (Robinson, 2000).

RTSs occur on a great variety of slope aspects. While some studies investigating different regions across the Arctic reported that their observed RTSs tended to have different prevailing slope orientations (Kokelj et al., 2009; Lacelle et al., 2015; Jones et al., 2019; Nesterova et al., 2021; Bernhard et al., 2022), several other studies found that higher RTS ablation rates and headwall retreat (see Sect. 3.1.1) are related to southern aspects (Lewkowicz, 1987a; Grom and Pollard, 2008; Lacelle et al., 2015). However, several other studies did not find any link between the slope aspect and RTS activity (Wang et al., 2009; Nesterova et al., 2021; Bernhard et al., 2022). Bernhard et al. (2022) suggested that differences in the RTS aspect may be explained by regional geological history that defines ice content and ice distribution, which are the main factors of RTS occurrence (Mackay, 1966; Kerfoot, 1969).

## 2.3. Ground ice

A high excess ground ice content is a prerequisite for RTS occurrence. The shallower the ground ice table the higher the likelihood that seasonal thawing will reach and start melting the ice, potentially triggering the initiation of the RTS. Regions with abundant ground ice presence in Canada feature widespread and ubiquitous slumps (Lamothe and St-Onge, 1961; Mackay, 1966; Kokelj et al., 2017). Similar observations were reported for Central Yamal, Russia (Babkina et al., 2019). RTS in areas with a thinner ground ice-rich layer tend to stabilize faster due to the rapid ice exhaustion (Kizyakov, 2005). The type of ground ice and its local distribution can define some morphologic characteristics of RTS (see Sect. 3.1) and affect retreat rates. For example, RTS forming in syngenetic ice-rich Yedoma deposits with polygonal ice wedges are usually accompanied by the presence of baydzherakhs (conical remnant mounds, for details, see Sect. 3.1.6) on the slump floors. De Krom and Pollard (1989) found that on Herschel Island, Canada, large ice wedges melted slower than the enclosing massive ground ice body. While abundant ground ice is necessary for RTS formation it is not the only control for RTS occurrence.

## 2.4. Triggers

An RTS forms once very ice-rich permafrost or massive ground ice becomes exposed for any reason and starts melting. Triggers of this exposure can be any anthropogenic or natural permafrost disturbances.

Anthropogenic triggers can be any disturbance of the thermal balance of the permafrost due to direct human actions. For example, mining was reported to trigger ice exposure and further formation of RTS in Canada (Fraser and Burn, 1997) and on the Qinghai–Tibetan Plateau, China (Wei et al., 2006; Niu et al., 2012).

Natural triggers can be separated into climatic, geomorphological, and wildfires. Wildfire removes vegetation and possibly the upper protective organic soil layer leading to deeper thaw than normal (Harry and MacInnes 1988; Jorgenson and Osterkamp,

2005; Lacelle et al., 2010). Climatic triggers are generally associated with a deepening of the active layer and the subsequent thawing of ice-rich deposits or massive ground ice. It can be caused by:

- unusually long warm weather periods (Lacelle et al., 2010; Balser et al., 2014; Swanson and Nolan, 2018; Lewkowicz and Way, 2019; Jones et al., 2019), or

- heavy precipitation and snowmelt events (Leibman et al., 2003; Lamoureux and Lafreniere, 2009; Balser et al., 2014).

Geomorphological triggers slightly differ inland and on the coasts. Coastal triggers reported in the literature include:

- thermal erosion (or thermo-erosion) (Burn and Lewkowicz, 1990; Lantuit et al., 2012; Kokelj and Jorgenson, 2013), or

- coastal erosion (Burn and Lewkowicz, 1990; Burn, 2000; Dallimore et al., 1996; Lantuit et al., 2012; Kokelj and Jorgenson, 2013; Ramage et al., 2017; Lewkowicz and Way, 2019).

RTS formation can also be initiated due to inland geomorphological triggers such as:

- development of thermo-erosional gullies downwards (Czudek and Demek, 1970; Jones et al., 2019),

- mechanical riverbank and lakeshore erosion (Burn, 2000; Burn and Lewkowicz, 1990; Lewkowicz and Way, 2019; Jones et al., 2019),

- thermokarst subsidence (Romanovskii, 1993; Voskresenskii, 2001),

- active layer detachment slides (Lewkowicz and Harris, 2005; Lacelle et al., 2010; Swanson, 2021; Jorgenson and Osterkamp, 2005; Lewkowicz, 2007; Lamoureux and Lafreniere, 2009; Lewkowicz and Way, 2019; Jones et al., 2019), or

- ice-wedge melt that leads to the degradation of ice-rich slopes (Fraser et al., 2018).

Two additional interesting but maybe highly site-specific inland geomorphological triggers were reported to be possible reasons for RTS re-initiation:

- the growth of a talik that occurs in ice-rich permafrost can lead to thaw subsidence and stimulate further RTS reoccurrence (Kokelj et al., 2009), or

- the growth of a debris tongue (thawed sediments in the shape of a tongue, for details, see Sect. 3.1.8) can eventually obstruct a stream valley and lead to the rise of stream base-level and further thermo-erosion that can erode and expose the ground ice and secondary RTS occurrence (Kokelj et al., 2015).

## 2.5. Polycyclicity

RTSs can develop in a polycyclic fashion, which means they can be active, then temporarily stabilize, and reactivate again (Mackay, 1966; Kerfoot, 1969; Kokelj et al., 2009). Yet some may end off in one cycle. RTSs can be considered active when there is an ongoing ablation of the exposed ice and thawed material is transferred downslope. Some studies reported continued headwall retreat and thawed sediment fluxes even in slumps where the ice was covered by the sediments (Kokelj et al., 2015; Zwieback et al., 2020). The reasons for these sediment-covered slumps to retain activity were heavy rainfalls and unsuppressed heat flux to the ice.

RTSs can stabilize mostly for two reasons: 1) exposed ground ice has completely melted, or 2) the exposed ice is re-buried by
sediments and thermally fully insulated from further melting (Burn and Friele, 1989). Once an RTS is stabilized, pioneer
vegetation starts to grow in the slump floor. Vegetation in stabilized RTS can go through several stages of succession and for
stabilized RTS in Yukon Territory, Canada, it was reported that forest and tundra communities were re-established after 35-
50 years (Burn and Friele, 1989). Some researchers found that RTSs can be stabilized for up to several hundred years in West
Siberia, Russia, (Leibman et al., 2014). Such long-term stabilized RTS are labeled in some studies as ancient (Nesterova et al.,

175    2023).

New active RTS can form within the outline of another stabilized RTS, moreover, neighboring RTSs can grow and coalesce
at some point (Lantuit and Pollard, 2008). This leads to the very complex spatial organization of nested and amalgamated
RTSs of sometimes different ages. It raises additional challenges when delineating and mapping RTS and their characteristics
(van der Sluijs et al., 2023; Leibman et al., 2023).
**2.6. Concurrent processes**
While triggering processes described in Sect. 3.2 takes place before RTS initiation, concurrent processes start simultaneously
or soon after RTS initiation and often are further reinforced by RTS growth. Depending on the terrain, concurrent processes
can have different impacts on RTS.
If RTS initiation occurs along the ice-wedge polygonal network, it also affects its further development (Fraser and Burn, 1997;
Fraser et al., 2018). Ice-wedge degradation can result in the rugged outline of the headwall following the morphology of
wedges (Fig.2a, b).
RTSs at the sea coast or any large water bodies (lakes, rivers) where wave action takes place are affected by coastal erosion at
the bluff base. This process manifests itself in washing away the debris tongue of RTS thus promoting further RTS growth by
removing debris blockages (Burn and Friele, 1989; Are, 1998), steepening the erosional base by coastal retreat, and in some
cases also undercutting the coast and niche-formation adding to further collapse of steep shore bluffs (Fig.2e).
As mentioned above, RTSs can form due to massive ground ice exposure in thermo-erosional gullies. Usually in such RTSs
lateral thermo-erosion continues to act simultaneously with ice ablation and thaw-related mass wasting. Sometimes lateral
thermo-erosion can appear in already existing RTS (Kerfoot, 1969; Lantuit and Pollard, 2005; Gubarkov et al., 2009). In both
cases, the RTS has or develops a specific gully-like shape (Fig.2c).
Due to specific RTS geometries and climatic conditions, thick snow packs accumulating from wind drift of snow in the winter
can remain within some RTS over summer (Fig.2d). This can affect RTS development by thermal insulation (Zwieback et al.,
2018). It was reported that snow packs prevented fast headwall retreat rates compared to the headwalls not covered by snow
(Lacelle et al., 2015).
Thermokarst subsidence and ponding can also occur within a slump floor (Fig.2b). This happens if the ground ice lies deeper
than the level of the slump floor, and there are conditions for water to accumulate in local concavities (Leibman and Kizyakov,
2007). It also can happen that meltwater streams can go into ice wedge tunnels, disappear in sinkholes on the slump floor, and
reappear further down at the slump floor.

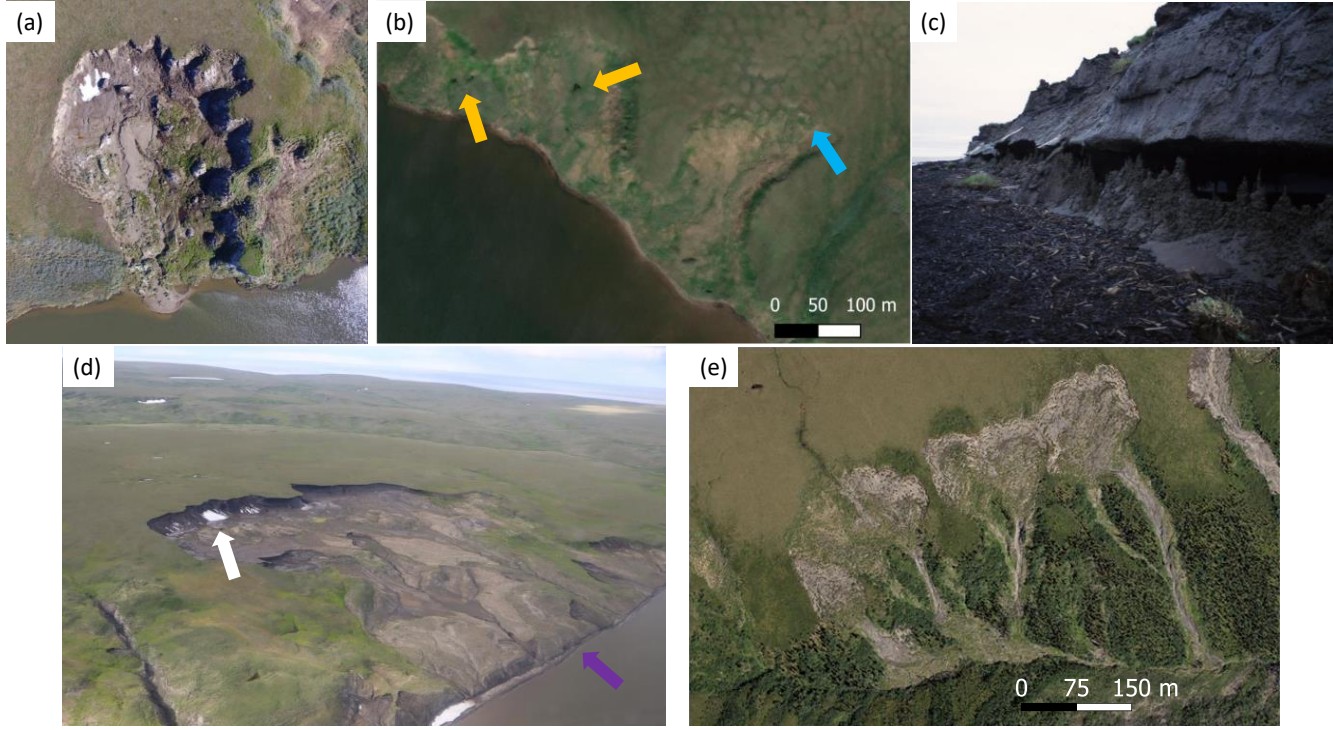

**Figure 2 Concurrent processes affecting RTS: (a) ice-wedge degradation in an RTS on Yamal Peninsula, West Siberia, Russia, July**
**2018, UAV photo: Artem Khomutov, (b) Thermokarst subsidence indicated by yellow arrows and ice-wedge degradation indicated**
**by a light-blue arrow in a stabilized RTS on Gydan Peninsula, West Siberia, Russia, July 2019, ESRI Basemap, GeoEye-1 satellite**
**image, (c) Erosional niche formed due to the coastal erosion affecting RTS, Oyagos Yar, NE Siberia, Russia, September 2002, photo:**
**Guido Grosse, (d) white arrow indicates snow packs staying over summer, the purple arrow indicates an area where coastal erosion**
**undercuts the coast and washes away debris tongue of an RTS on Herschel Island, Northern Canada, July 2022, photo from**
**helicopter: Saskia Eppinger, (e) active thermal erosion in RTSs that occurred within gullies near Willow River, NW Canada, July**
**2023, aerial camera image, credit: AWI. ESRI basemap used in (b) has the following credits: Esri, DigitalGlobe, GeoEye, i-cubed,**
**USDA FSA, USGS, AEX, Getmapping, Aerogrid, IGN, IGP, swisstopo, and the GIS User Community.**

**3 Terminologies used in the literature**
**3.1. Morphologic parts**
RTS have various morphologic parts, of which some are characteristic of all RTS but some may be present only in certain RTS
types and depend on local geological conditions (Figure 3). Moreover, some of these RTS features can still be visible even
after the RTS stabilized. Some studies use various terms to describe the same parts of RTS, which then are synonymous terms,
while other studies use the same terms but actually describe partially or fully different parts of the RTS with them. This can
cause confusion when comparing RTS characteristics across different studies (Table 1).

**Table 1 Morphologic parts of RTS and different terminologies used to describe them. The last column represents the presence "+" or the absence "-" of the morphologic part in stabilized RTS.**

| Most common term | Other related terms | Description | Presence in stabilized RTS |
|---|---|---|---|
| **Present in all RTS (essential)** | | | |
| 1. *Headwall* (Kerfoot, 1969; Egginton, 1976; Burn and Friele, 1989; Burn and Lewkowicz, 1990; Barry, 1992) | • *Backwall* (Lamothe and St-Onge, 1961; Worsley, 1999; Leibman et al., 2021)<br>• *Headscarp* (De Krom, 1990; Lewkowicz, 1987b; Lantuit and Pollard, 2005)<br>• *Slump face* (Huang et al., 2022)<br>• *Ice face* (Kerfoot, 1969; Lewkowicz, 1987b)<br>• *Scarp* (Mackay, 1966; Kerfoot, 1969; Egginton, 1976; Fortier et al., 2007; Wang et al., 2009; Nicu et al., 2021)<br>• *Escarpment* (Swanson and Nolan, 2018; Swanson, 2021) | A steep retreating wall consisting of ablating ice and frozen sediments at the back of RTS | - |
| 2. *Slump floor* (Mackay, 1966; Lewkowicz, 1987a; Burn and Friele, 1989; De Krom, 1990; Barry, 1992; Lantuit and Pollard, 2005; Lacelle et al., 2010)<br>or<br>*Scar* (De Krom, 1990; Barry, 1992; Kokelj et al., 2002; Kokelj et al., 2009) | - | The low-angle to horizontal area of the hollow's bottom | + |

| | | | |
|---|---|---|---|
| 3. *Mudflow* (Lamothe and St-Onge, 1961; Egginton, 1976; Lewkowicz, 1987a) | • *Earth / Mud flow* (Leibman et al., 2014)<br>• *Debris flow* (Murton, 2001; Lipovsky and Huscroft, 2006) | The meltwater stream that carries thawed viscous sediment material downslope across and out of the slump floor | - |
| 4. *Edge* (Cassidy et al., 2017; Leibman et al., 2021; Leibman et al., 2023; van der Sluijs et al., 2023; Kizyakov et al., 2023) | • *Outline* (Burn, 2000; Yang et al., 2023) | The boundary line of the headwall or entire landform | + |
| **Present in some RTS depending on various local characteristics (optional)** | | | |
| 5. *Mudpool* (De Krom and Pollard, 1989; Lantuit and Pollard, 2005) | - | The area of the first accumulation of thawed liquid material, generally at the base of the headwall | - |
| 6. *Evacuation channel* (Lacelle et al., 2004; Lacelle et al., 2010; Delaney, 2015) | - | Channel the thawed sediments and meltwater (debris) pass through when leaving the slump floor | + |
| 7. *Debris tongue* (Worsley, 1999; Kokelj et al., 2015; Segal et al., 2016) | • *Slump lobe* (Lantuit and Pollard, 2005)<br>• *Mud lobe* (Lantuit and Pollard, 2005) | Thawed sediments and meltwater (debris) in the shape of a tongue that slid downslope from the slump floor | + |
| 8. *Slump block(s)* (Swanson, 2012; Kokelj et al., 2015) | • *Remnant island* (Burn and Friele, 1989; Bartleman et al., 2001) | Pieces of soil and vegetation that slid or fell from the headwall and are located within a slump floor | - |
| 9. *Baydzherakh(s)* (Czudek and Demek 1970; Zhigarev, 1975; Pizhankova, 2011; Séjourné et al., 2015) | - | Conical hills within a slump floor remnant after thawing of large ice-wedges | + |
| 10. *Mud levees* (Kerfoot, 1969; Lantuit and Pollard, 2005) | - | "Dams" of dried stagnated thawed sediments within a slump floor | - |

| | | | |
|---|---|---|---|
| 11. *Mud gullies* (Lantuit and Pollard, 2005) | - | Erosional channels within thawed sediments formed by meltwater flow within a slump floor | **-** |
| 12. *Dropwall* (Leibman et al., 2021) | - | A cliff between the edge of the hanging RTS floor and the shore | + |
| 13. *Side-wall* (Lewkowicz, 1987b) | - | A steep retreating wall consisting of ablating ice at the side of RTS | **-** |

### 3.1.1. Headwall and Side-walls

The term *headwall* is used in the literature in two ways: 1) as a broad general term for the steep wall of RTSs, where the ice is exposed (Kerfoot, 1969; Egginton, 1976; Burn and Friele, 1989; Burn and Lewkowicz, 1990; Barry, 1992) and 2) as a term for only the upper vertical part of the wall that consists of the active layer and ice-poor organic or mineral sediments (Lantuit and Pollard, 2005; Lewkowicz and Way, 2019). The second lower part of the RTS wall according to these authors is a steep (20°-50°) *headscarp* that consists of exposed ice-rich sediment or massive ground ice. Exposed ice is not only called a *headscarp* in the literature but sometimes also an *ice face* and in such cases, the *ice face* is a part of the headwall that represents the whole RTS wall in a general way (Kerfoot, 1969; De Krom, 1990; Burn and Lewkowicz, 1990; Barry, 1992).

There are several terms in the literature that are used to describe the whole RTS wall (*headwall* in a general way): for example, *slump face* (Huang et al., 2022), *scarp* (Mackay, 1966; Kerfoot, 1969; Egginton, 1976; Fortier et al., 2007; Wang et al., 2009; Nicu et al., 2021) and *escarpment* (Swanson and Nolan, 2018; Swanson, 2021). Another similar term is a *backwall* and it is used to describe the whole RTS wall but separate it by its location on the back of the RTS (Lamothe and St-Onge, 1961; Worsley, 1999; Leibman et al., 2008). Those RTS walls that are located at the sides are sometimes called *side-walls* (Lewkowicz, 1987b). Side-walls can be called an optional morphologic part since they mostly occur only in bowl-shape morphologies.

Since a *headwall* is a wall with exposed ablating ice and frozen sediments, it can only be found in an active RTS. The remnants of the headwall in stabilized RTSs are sometimes called *stable headwall* (Kokelj et al., 2009) or *old headscarp* (Zwieback et al., 2018).

### 3.1.2. Slump floor or Scar

As a *headwall* retreats it leaves a low-angle surface that can also be described as the bottom of the RTS hollow. This surface is termed *slump floor* (Mackay, 1966; Lewkowicz, 1987a; Burn and Friele, 1989; De Krom, 1990; Barry, 1992; Lantuit and Pollard, 2005; Lacelle et al., 2010), highlighting its flatness or sometimes with the term *scar* (De Krom, 1990; Barry, 1992;

Kokelj et al., 2002; Kokelj et al., 2009) that originates from landslide terminology and means the bare surface that is left after the removal of the mobilized sediments by mass movement. Both of the terms are equally popular in the literature and sometimes are used simultaneously in the same study as an interchangeable term (De Krom, 1990; Barry, 1992). A *slump floor* or *scar* can be found in active as well as stabilized RTSs.

### 3.1.3. Mudpool and Mudflows

The area of the mud in the *slump floor* right next to the headwall is often (but not always) the place where meltwater accumulates. Some authors call this area of the RTS slump floor a *mudpool* (De Krom and Pollard, 1989; Lantuit and Pollard, 2005). Thawed sediments after their first accumulation in the *mudpool* are transported downslope by the streams of meltwater. These flows of meltwater-saturated mud depending on the amount of water are generally called *mudflows* (Lamothe and St-Onge, 1961; Egginton, 1976; Lewkowicz, 1987a), *earth/mud flows* (Leibman et al., 2014) and *debris flows* (Murton, 2001; Lipovsky and Huscroft, 2006).

### 3.1.4. Mud gullies and levees

Meltwater streams can lead to the formation of *mud gullies* within a *slump floor* – erosional channels that are carved by meltwater streams into debris (Lantuit and Pollard, 2005). If transported debris stagnates and dries out it may form *mud levees* bordering *mudflows* (Kerfoot, 1969; Lantuit and Pollard, 2005).

### 3.1.5. Slump block

The pieces of ice-poor, often organic-rich peaty soil covered with vegetation that slide down the headwall into the slump floor and stay rigid when moving downslope with mudflows are called *slump blocks* in some studies (Swanson, 2012; Kokelj et al., 2015). If these features consist of active layer soil, they generally preserve the initial undisturbed tundra vegetation. Some authors called these blocks also *remnant islands* (Burn and Friele, 1989; Bartleman et al., 2001).

### 3.1.6. Baydzherakh(s)

*Baydzherakhs* (from the Yakutian language, but now a more commonly accepted term) are conical mounds in the *slump floor* of RTSs representing largely still frozen remnants of ice-wedge polygon centers where the surrounding polygonal large ice wedges have thawed substantially already. They are typical for RTSs located on upland slopes with ice-rich deposits and large polygonal ice wedges up to 50 m thick (i.e., Yedoma Ice Complex) (Tikhomirov, 1958; Czudek and Demek, 1970; Zhigarev, 1975; Pizhankova, 2011; Séjourné et al., 2015). *Baydzherakhs* can reach significant sizes: up to 11 m in height, 15 m in width, and 20 m in length (Tikhomirov, 1958). Thus, they can be found not only in active but also in stabilized RTSs. As a typical feature of Yedoma upland slopes *baydzherakhs* are widely distributed in the Yedoma Ice Complex regions of Eastern and North-Eastern Siberia, Alaska, and North-Western Canada (Strauss et al., 2021) as well as in other areas formed by ice-rich

deposits with large polygonal ice wedges. *Baydzherakhs* will therefore not form in areas where RTSs are formed in deposits with thick ice layers.

### 3.1.7. Evacuation channel

Depending on the morphology of an RTS, thawed sediments and meltwater (debris) can leave the *slump floor* through the trench connecting the *slump floor* and the base level. This optional morphologic part of RTSs is termed an *evacuation channel* (Lacelle et al., 2004; Lacelle et al., 2010; Delaney, 2015).

### 3.1.8. Debris tongue

Thawed sediments and meltwater (debris) moving downslope can eventually escape from the *slump floor* directly or via an *evacuation channel*. Once this happens, thawed sediments accumulate in the shape of a "tongue" on any surface where an RTS outflow ends. Such features are generally called *debris tongues* (Worsley, 1999; Kokelj et al., 2015; Segal et al., 2016), but are sometimes referred to as *mud* or *slump lobes* (Lantuit and Pollard, 2005).

### 3.1.9. Edge and dropwall

The term *edge* of RTS is used in the literature to indicate: 1) the outline of the whole feature (van der Sluijs et al., 2023) and 2) the boundary line of active retreat (Cassidy et al., 2017; Leibman et al., 2021; Leibman et al., 2023; Kizyakov et al., 2023). There is also the term *outline* itself that is used to describe the whole area of the RTS landform (Burn, 2000) or only the polygon that is considered to be the RTS detected by automated mapping methods (Yang et al., 2023). Furthermore, the *edge* of RTS is also sometimes classified into upper edge meaning the boundary line of active retreat of the *headwall* (Kizyakov et al., 2023), and *lower edge* meaning the boundary line of the cliff retreat for RTSs on the sea coasts (Leibman et al., 2008; Leibman et al., 2021). The face (cliff) from the *lower edge* of coastal RTS to the beach level has been called a *dropwall* (Leibman et al., 2021) to differentiate this morphologic part of the RTS from the rest of the coastal cliff.

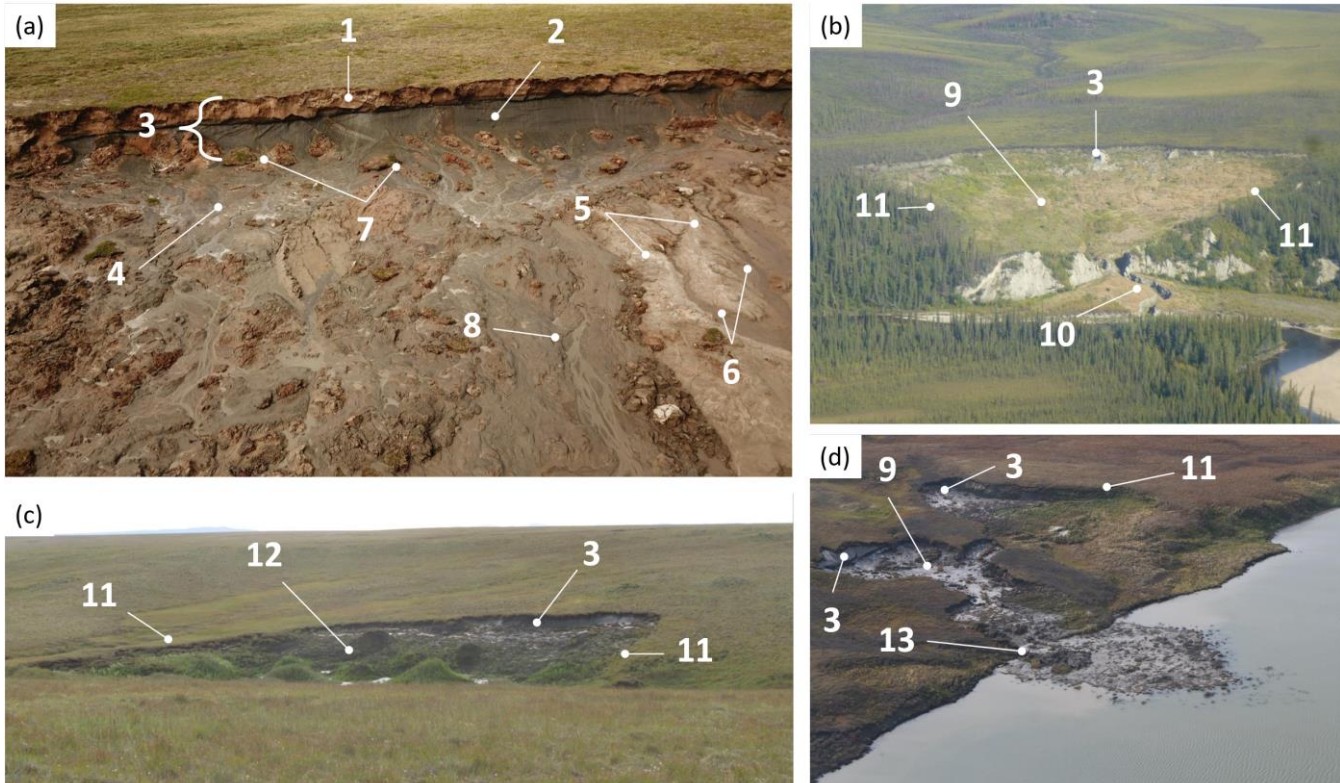

293

**Figure 3 Morphologic parts of active RTSs in (a) Yamal Peninsula, West Siberia, Russia, July 2019, unmanned aerial vehicle (UAV) photo: Nina Nesterova, (b) Alaska, USA, August 2016, photo from the airplane: Ingmar Nitze, (c) Bykovsky Peninsula, NE Siberia, Russia, August 2015, field photo: Alexandra Veremeeva, (d) Gydan Peninsula, West Siberia, Russia, 2020, photo from helicopter: Elena Babkina. The numbers on the photos stand for the following morphologic parts: 1 – headwall, i.e. the upper vertical part of the wall only; 2 – headscarp; 3 – headwall (or a backwall), more generally describing the entire steep wall; 4 – mudpool; 5 – mud levees; 6 – mud gullies; 7 – slump block; 8 – mudflow; 9 – slump floor or scar; 10 – evacuation channel; 11 – side-wall; 12 – baydzherakhs; 13 – debris tongue.**

## 3.2. Landforms

### 3.2.1. Retrogressive thaw slump (RTS)

According to the International Permafrost Association Multi-Language Glossary of Permafrost and Related Ground-Ice Terms (van Everdingen, 2005), an RTS is defined as: "A slope failure resulting from thawing of ice-rich permafrost. Retrogressive thaw slumps consist of a steep headwall that retreats in a retrogressive fashion due to thawing and a debris flow formed by the mixture of thawed sediment and meltwater that slides down the face of the headwall and flows away. Such slumps are common in ice-rich glaciolacustrine sediments and fine-grained diamictons."

### 3.2.2. Cryogenic earthflow

In Russian literature, the word *cryogenic* is usually used to describe the periglacial nature of the processes. It refers to thermophysical, physicochemical, and physicomechanical processes occurring in freezing, frozen, and thawing deposits (van Everdingen, 2005). This term is usually omitted in the literature in English (Poppe and Brown, 1976).

The term *cryogenic earthflow* was introduced by Leibman (1997, in Russian) meaning a viscous or viscoelastic flow of water-saturated soil of the active layer sliding on the surface of massive ground ice bodies or the table of ice-rich permafrost. The examples of cryogenic earthflows in Central Yamal are demonstrated in Fig.4.

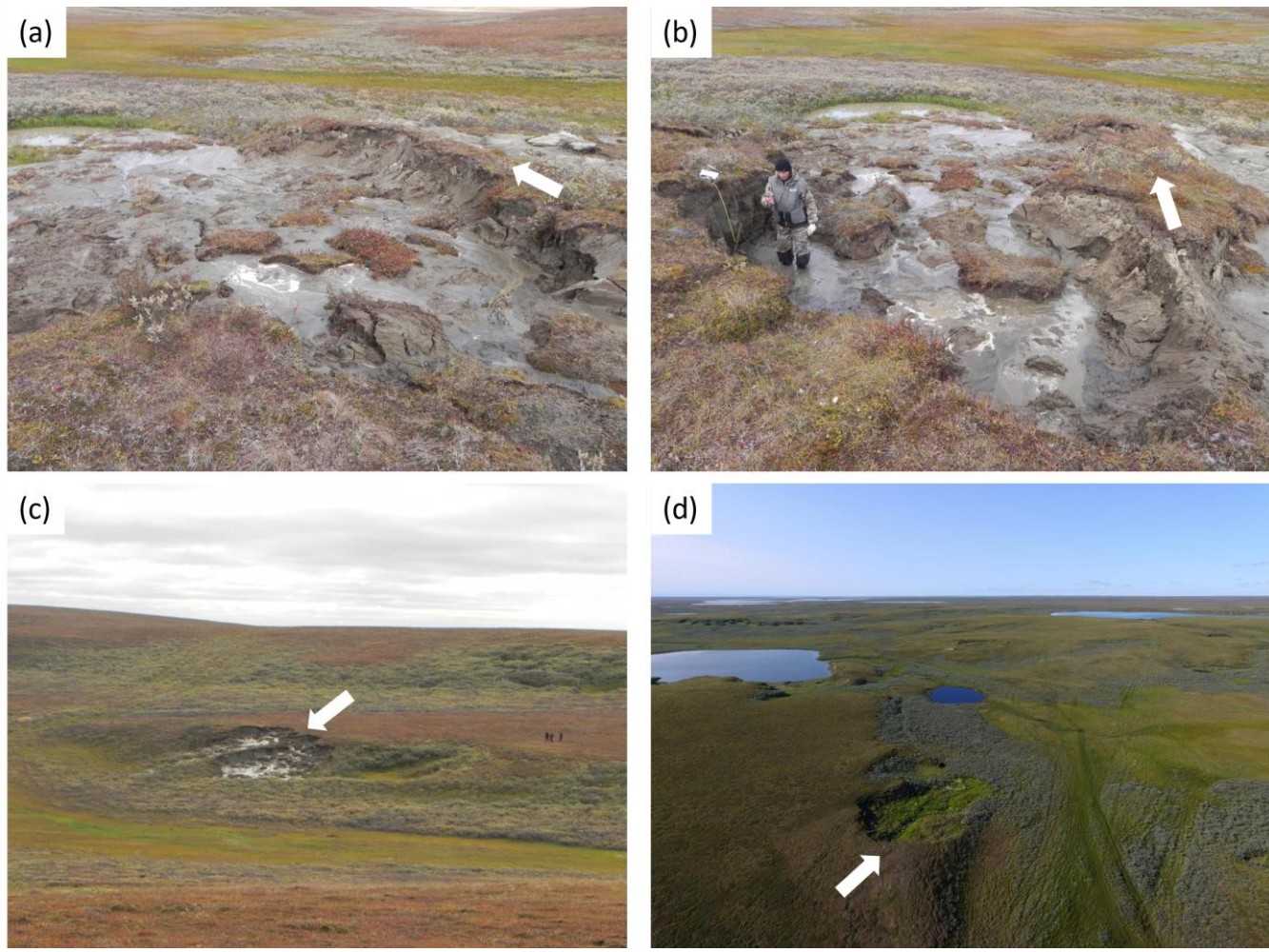

**Figure 4 The examples of two cryogenic earthflows next to each other being active on (a), (b) and (c) photos in Central Yamal, West Siberia, Russia made on a photo camera in 2012 and stabilized on (d) the photo made by UAV in 2017. The arrow indicates the direction of flow. Photos: Artem Khomutov.**

### 3.2.3. Thermocirque

The term *thermocirque* was first mentioned by Czudek and Demek (1970, in English) to describe "amphitheatrical hollows" that occur after ice wedge melt in the gullies at the river banks in Yakutia (Russia). Thermocirques according to the authors had "a vertical and overhanging slope at the head and an uneven floor". In Russian-language literature, the term thermocirque was sometimes called by interchangeable term "*thermokar*" when describing a round or cirque-like hollow at the river banks or the lake shores composed of icy permafrost (Grigoriev and Karpov, 1982, in Russian; Voskresenskii, 2001, in Russian). Following the development of theoretical concepts of cryogenic landsliding (Sect. 3.2.3 and 3.2.4) the term thermocirque was defined as an extensive landform resulting from a series of multi-aged cryogenic earthflows (Leibman, 2005, in Russian; Leibman et al., 2014). The scheme visualizing thermocirque formation and the example of the thermocirque in Central Yamal, Russia are demonstrated in Fig.5.

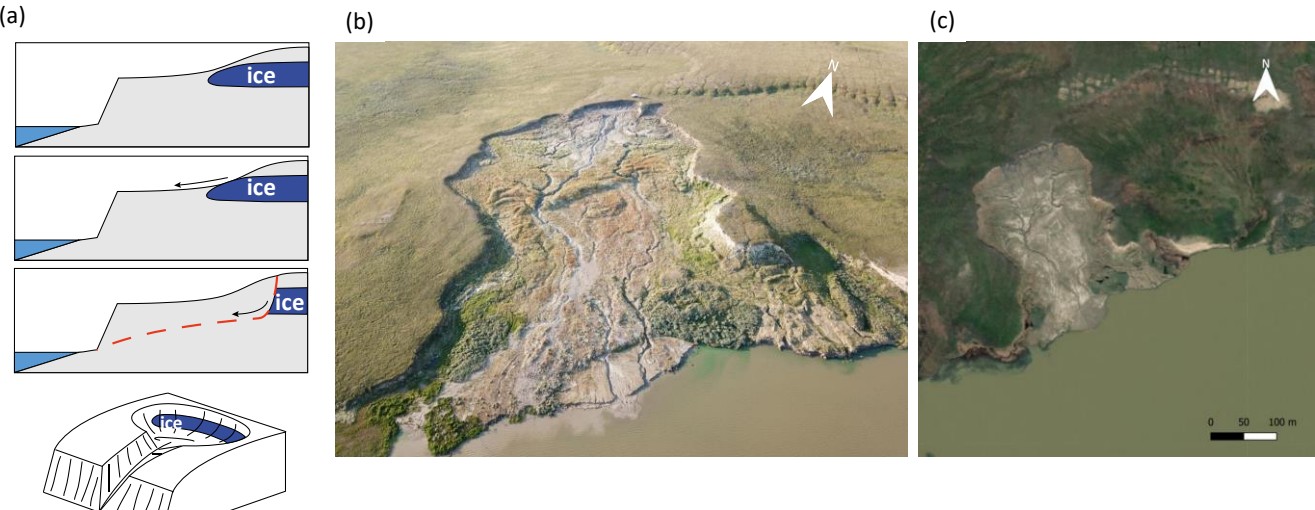

**Figure 5 (a) Scheme of Thermocirque formation, the black arrow indicates the direction of mass movement of thawed material, and the red dashed line stands for the cross-section of the headwall and the slump floor (see Sect. 3.5). Note that the scheme demonstrates the particular ice morphology of a layer of tabular massive ground ice (adapted from Kizyakov, 2005), but may also consist of other forms of ice-rich ground. Example of a thermocirque in Central Yamal, West Siberia, Russia show on (b) in a UAV photo in August 2019 (photo: Artem Khomutov) and (c) in a WorldView-2 satellite image from July 2018 (Source: ESRI satellite basemap). ESRI basemap used in (c) has the following credits: Esri, DigitalGlobe, GeoEye, i-cubed, USDA FSA, USGS, AEX, Getmapping, Aerogrid, IGN, IGP, swisstopo, and the GIS User Community.**

### 3.2.4. Thermoterrace

The term *thermoterrace* was first mentioned by Ermolaev (1932, in Russian) to describe "picturesque outcrops of ice falling vertically onto a narrow, 1-2 m wide space located along the seashore along the edge of the ice wall that can reach 30-35 m". The local term to describe these icy cliffs was muus kygams - муус кьham in Yakutian language (Ermolaev, 1932). The more

precise definition of thermoterrace was given by Zenkovich and Popov (1980) as a terrace-like area in the upper part of the icy cliff at the seashore that results from the cliff retreat due to the thermal influence of warm air and solar radiation. Thermoterraces were reported to reach up to a few km in length along the coast and more than 200 m in width (Are et al., 2005). A scheme visualizing thermoterrace formation based on Kizyakov (2005) and an example of a thermoterrace on the Bykovsky Peninsula, Yakutia, Russia are shown in Fig.6.

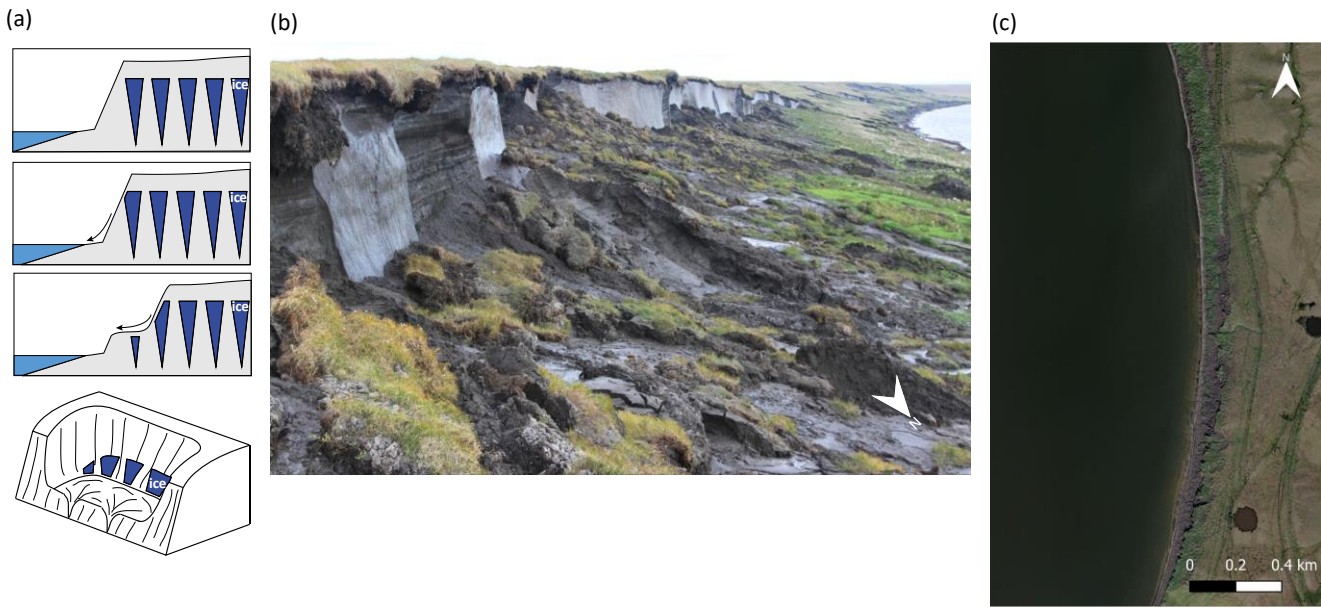

**Figure 6 (a) Scheme of thermoterrace formation, the black arrow indicates the direction of mass movement of thawed material; note that the scheme demonstrates the particular ground ice morphology of a layer with large ice-wedges (adapted from Kizyakov, 2005), but may also consist of other ground ice morphologies. Example of Thermoterraces in Bykovsky Peninsula, NE Siberia, Russia shown (b) on the ground in August 2016 (photo: Alexander Kizyakov) and (c) in a WorldView-2 satellite image from August 2020 (ESRI satellite basemap). ESRI basemap used in (c) has the following credits: Esri, DigitalGlobe, GeoEye, i-cubed, USDA FSA, USGS, AEX, Getmapping, Aerogrid, IGN, IGP, swisstopo, and the GIS User Community.**

### 3.2.5. Active layer detachment slide

Another closely related slope landform linked to RTS formation (see Sect. 3.2) is an active layer detachment slide or failure (ALD). The term ALD is prevalent in recent publications (Blais-Stevens et al., 2015; Balser, 2015), yet, unlike RTS, there is no universally endorsed term to describe ALD phenomena in the Glossary (van Everdingen, 2005):

• active layer failure - "A general term referring to several forms of slope failures or failure mechanisms commonly occurring in the active layer overlying permafrost" (not recommended synonym: skinflow)

• detachment failure - "A slope failure in which the thawed or thawing portion of the active layer detaches from the underlying frozen material" (not recommended synonyms:  skin flow, active layer glide)

French (2018) defines active layer detachment slides as rapid slope failures restricted to the active layer that generally occur at the middle or upper slopes. In one of several classical works on ALD, Lewkowicz (1990) defines ALD as being a rapid

mass movement on permafrost slopes without strict limitation to the active layer: "Failure involves the unfrozen mass detaching from the underlying substrate and sliding downslope over a thawing ice-rich zone within the active layer or the upper part of the permafrost." Examples of mass movements that seem deeper than the active layer and are nevertheless termed ALD are shown by Rudy et al. (2016).

### 3.2.6. Cryogenic translational landslide

The term *cryogenic translational landslide* (CTL) was suggested by Kaplina (1965, in Russian), and the definition was later elaborated in further publications based on observations in Central Yamal, Russia (Leibman and Egorov, 1996; Leibman, 1997; Leibman et al., 2014). The definition of CTL summarized from the abovementioned publications can be phrased as single-time lateral displacement of thawed soil block sliding on the surface of the seasonal ice formed at the active layer base. This type of seasonal ice is formed due to the active layer's upward freezing, ice aggradation at the base of the active layer, and later melting (Leibman et al., 2014; Lewkowicz, 1990). Examples of CTL in Central Yamal are shown in Fig.7.

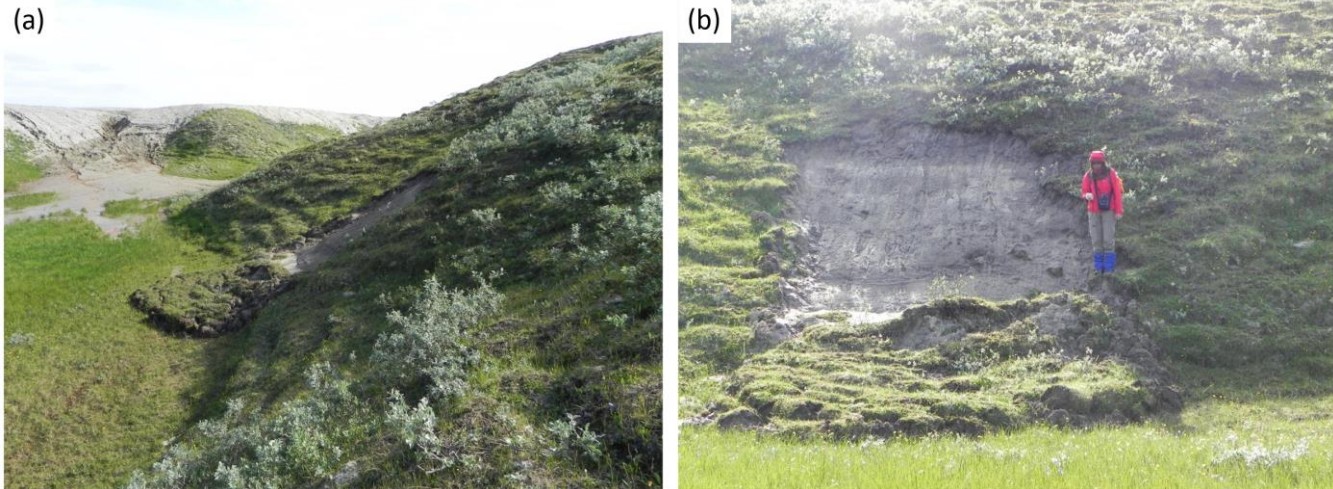

**Figure 7 Example of a cryogenic translational landslide (CTL) in 2019 in Central Yamal, West Siberia, Russia (photo: Artem Khomutov), (a) view from the side and (b) view from the front.**

### 3.3. Formation process

The process of RTS formation in the recent literature is termed in two different ways: as *thermokarst* and as *thermodenudation*.

### 3.3.1. Thermokarst

The term *thermokarst* was first suggested by Ermolaev (1932) to describe the surface subsidence due to the melting of ground ice as a similarity to the *karst* process by dissolution. However, in the context of RTS formation processes the term *thermokarst* is mostly referred to in the North American literature as a set of processes that lead to the occurrence of specific landforms due to the thawing of ice-rich permafrost or melting of massive ground ice (Kokelj and Jorgenson, 2013).

### 3.3.2. Thermodenudation

The term *thermodenudation* originally was suggested by Panov (1936), defining "the influence of the sun in a direct or transformed form through soil or water on the sediments containing a certain amount of ice cement or ice masses, as well as on bedrock with negative temperature <...> that leads to mass-wasting as well as some forms of thermo-erosion or thermokarst". In the context of RTS formation, this term has been used referring to ground ice thaw and slope mass waste (Leibman et al., 2021) as well as the retreat of upper bluff edges along coastal RTS (Guenther et al., 2012).

## 4 Discussion

### 4.1 Divergent terminologies

The terminology used to describe the RTS formation processes and related landforms in 21st-century publications has historical roots in the distinct scientific approaches developed in the USSR and North America (both Canada and the USA) during the 20th century.

The process of RTS formation following the initiation by various triggers and the further development into the landform has neither been named nor specifically classified in classical works on RTS and exposed ground ice (Mackay, 1966; Mackay, 1970; Rampton and Mackay, 1971; Lewkowicz, 1987a; Burn and Friele, 1989).

In the literature of the 20th century, this process was often termed *solifluction, thermokarst*, and *thermodenudation*. Initially, none of these three terms took the more specific formation of RTS into account in their definitions. At some point, however, the definitions of these three terms were expanded to include RTS formation. The process of RTS formation was also previously very broadly referred to as the process of *erosion* (Lamothe and St-Onge, 1961), but this term was later no longer used in publications in this context.

The general chronology of usage of these three terms which differ in definitions in the 20th century is shown in Fig.8. While this chronology graph has some limitations due to the a) ambiguity of some definitions, b) definition reformulation by some authors through their later publications, and c) usage of several terms for the same process etc., it helps understanding how the RTS terminology evolved in the scientific literature and how different schools of thought influenced its development.

The term *solifluction* was first introduced by Andersson (1906) and describes the process of slow downslope movement of saturated unfrozen materials (van Everdingen, 2005). In non-Russian language literature, this term has always been used for very slow movements up to several centimeters per year (Smith, 1988) and never for the rapid mass-wasting that can lead to RTS. Meanwhile, Russian-language authors have included the process of slumping into the *solifluction* calling it *rapid solifluction*. Probably the most remarkable publication with such a statement was issued by Kaplina (1965). The concept of rapid solifluction was later criticized by Dylik (1967) and Leibman (1997) for summarizing processes that have process rates differing by several orders of magnitude under one term. Nevertheless, this approach of referring to the RTS formation process

as *rapid solifluction* was frequently used in the literature until the end of the 20th century. The last publication in which *rapid solifluction* was mentioned in connection with the formation of RTS was by Yershov (1998).

In general, the term *thermokarst* has mostly been used by Russian-language researchers for describing the subsidence of the land surface (Sumgin et al., 1940; Kachurin, 1955; Mukhin, 1960; Dostovalov and Kudryavcev, 1967; Shur, 1977; Romanovskii, 1993; and many more later). Some exceptions can be found in two publications of Popov: one in English (Popov et al., 1966), where he included the slumping process in *thermokarst*, and another one in French (Popov, 1956), where his definition of *thermokarst* was not purely limited to the process of subsidence. Meanwhile, a different approach was suggested by Czudek and Demek (1970), who put the RTS formation process under the umbrella of the thermokarst term. They proposed two types of *thermokarst*: down-wearing which included only subsidence and back-wearing which included the RTS formation. This approach found support from French (1976), who extended this term by adding *thermal erosion* to it. French's (1976) definition of *thermal erosion* as "a dynamic process 'wearing away' by thermal means, i.e. melting of ice" differs from the one in the Glossary, where the main erosional agent is moving water: "The erosion of ice-rich permafrost by the combined thermal and mechanical action of moving water." This is the reason why the RTS formation process is sometimes called *thermal erosion*. For example, Burn (1983) relates the process of RTS formation to *thermal erosion*, which he in turn describes as part of the *thermokarst* process.

Since French (1976) expanded the definition of thermokarst processes to encompass slope processes and in particular thaw slumping, the RTS formation process has consistently been perceived as a thermokarst process in the North American literature (Washburn, 1979; Burn, 1983) or sometimes specified as hillslope thermokarst (Gooseff et al., 2009). There was no agreement among scholars on the terminology of the RTSs itself. RTSs were termed in the literature as tundra mudflows (Lamothe and St-Onge, 1961), ground-ice slumps (Mackay, 1966; French, 1976), retrogressive-thaw flow slides (Hughes, 1972), bi-modal flows (McRoberts and Morgenstern, 1974), or just thaw slumps (Washburn, 1979). The 1998 Glossary (van Everdingen, 2005) initially recommended using the term "retrogressive thaw slump", though alternative terms persist in later literature, such as "retrogressive thaw flowslides (thawslides)" (Wolfe et al., 2001) or "retrogressive thaw flows" (Highland and Bobrowsky, 2008).

Unlike RTS, the process of ALD was not always classified as thermokarst in the North American literature (Lewkowicz, 1990; Lewkowicz and Harris, 2005, etc.). For example, French (1976; 2018) describes ALD under the section of "Rapid mass movements", but not "Thermokarst" in all of the editions of his textbook "The Periglacial Environment". ALDs are included in the list of thermokarst processes described for Alaska by Jorgenson et al. (2008) and classified as hillslope thermokarst by Gooseff et al. (2009). Recent publications tend to include the process of ALD under the concept of thermokarst (Kokelj and Jorgenson, 2013; Ramage et al., 2019; Kokelj et al., 2023).

Additional definitions of thaw-related slope processes in the North American literature worthy of mention can be found in "The Landslide Handbook — A Guide to Understanding Landslides" of the United States Geological Survey by Highland and Bobrovsky (2008). In the section "Flows in permafrost", the authors define ALD as the "rapid flow of shallow layer of saturated

soil and vegetation" that moves over but on the underlying permafrost. Retrogressive thaw flows (as an analog term for RTS)
are described as the features resulting from the thawing of exposed buried ice lenses (Highland and Bobrowsky, 2008).

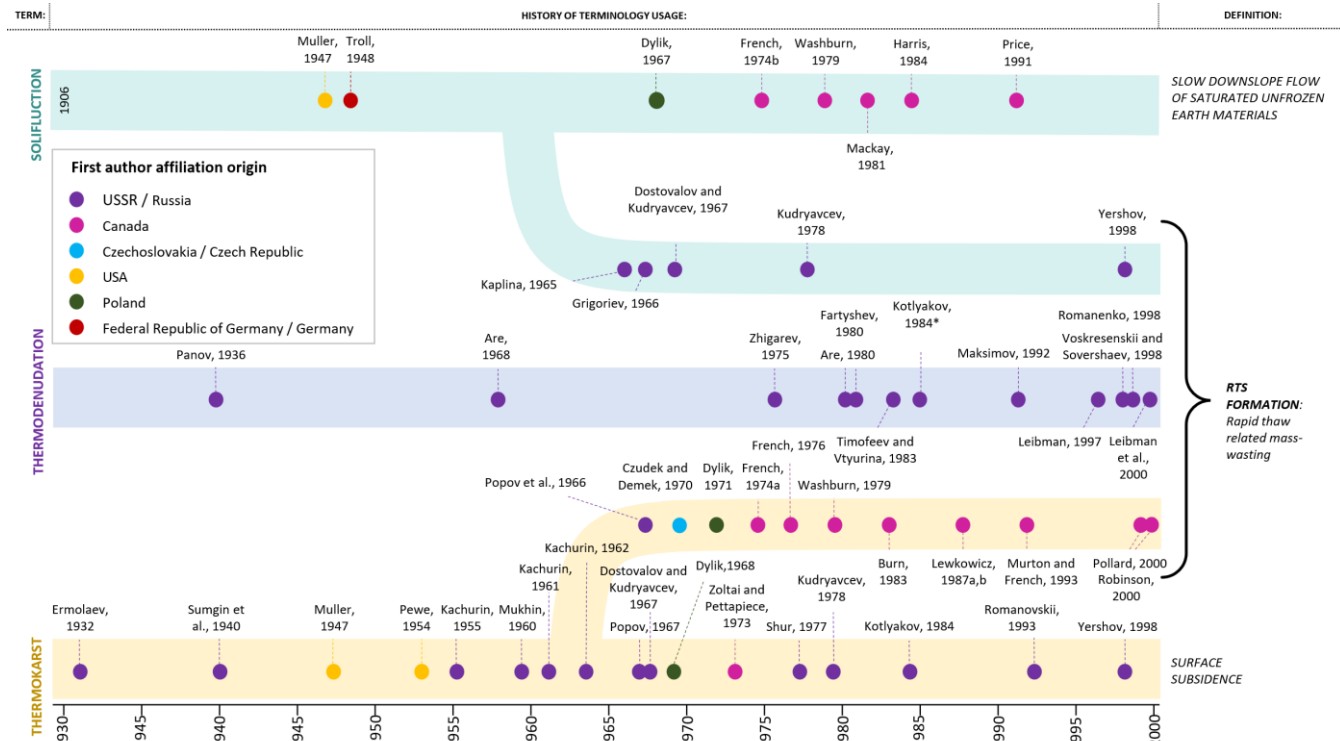


**Figure 8 Chronology of the usage of different RTS-related terms by selected most cited authors in the 20th century. Three color-coded wide lines represent the term on the left side and the main process by definition on the right side. The dots represent publications and are color-coded based on the first author affiliation origin.**

The term *thermodenudation* (sometimes also spelled as *thermal denudation*) has never been properly introduced in English-
language literature, however, it is widely used in Russian-language permafrost literature with two types of definitions: narrow
and wide. For the initial (narrow) definition see Sect. 3.3.2. Are (1968) used this term to describe the thermal effect of solar
radiation and sensible heat affecting the retreating ice-rich coastal cliffs. Zhigarev (1975) highlighted the importance of the
slope in his definition of thermodenudation as "a complex of gravitational and erosive processes that develop on slopes during
thawing of ice-rich deposits of various genesis". The only wide definition of *thermodenudation* was introduced in the Glossary
of Glaciology (Kotlyakov, 1984) as: "a set of cryogenic destructive processes and the transfer of the products of destruction
downwards. *Thermodenudation* includes cryogenic weathering, nivation, cryogenic slope processes (mass movements),
thermal erosion, thermal coastal erosion, thermokarst, and thermal suffosion". This wide definition by Kotlyakov (1984) of a
thaw-related process is quite similar to the expanded version of the thermokarst term by French (1976).
In the context of RTS formation, the term *thermodenudation* was widely applied in its narrow definition as a set of slope
processes associated with thawing of ice-rich deposits and leading to the occurrence of mass movements and concavities of

different shapes (Fartyshev, 1980; Romanenko, 1998; Leibman et al., 2021; and many more). These mass movements were classified by Leibman (1997) into two types depending on the sliding surface: cryogenic translational landslides on the seasonal ice in the base of the active layer (for detailed definition see Sect. 3.2.3) and cryogenic earthflows on the massive ice or icy permafrost (for detailed definition see Sect. 3.2.4).

Figure 9 demonstrates a conceptual scheme that explains the interrelation of different processes and lists the landforms resulting from *thermodenudation* (in narrow definition) in the Russian literature. When the cryogenic translational landslides do not lead to the exposure of ice-rich permafrost or massive ground ice, the surface of exposed bare soil gets revegetated (Khomutov and Leibman, 2016). Otherwise, if the icy deposits or the massive ice body are exposed because of this disturbance, a *cryogenic earthflow* can occur. Such features can also stabilize if the accumulation of drying sediments insulates the exposed ice. Once further thawing is suspended, the surfaces of these landforms get revegetated (Fig. 9) (Leibman, 2005). In contrast, the continued expansion of the flow and mass movements in several directions involving additional cryogenic processes lead to the formation of mature landforms defined in the literature as *thermocirque* (for detailed definition see Sect. 3.2.5) and *thermoterrace* (for detailed definition see Sect. 3.2.6). *Thermocirques* are reported in the literature to exhibit amphitheater-like shapes (Leibman et al., 2014), while thermoterraces are described as landforms elongated along the coast or the shore with coastal erosion contributing to the cut of its lower part (Are et al., 2005). The combinations of these two landforms are also observed in some regions (Leibman et al., 2023) (Fig. 10). This is usually found in two settings: one or more *thermocirques* form and grow in a former but now stabilized *thermoterrace* (Fig.10a), or when an originally separate *thermocirque* and *thermoterrace* merge at the coast into one outline (Fig.10b).

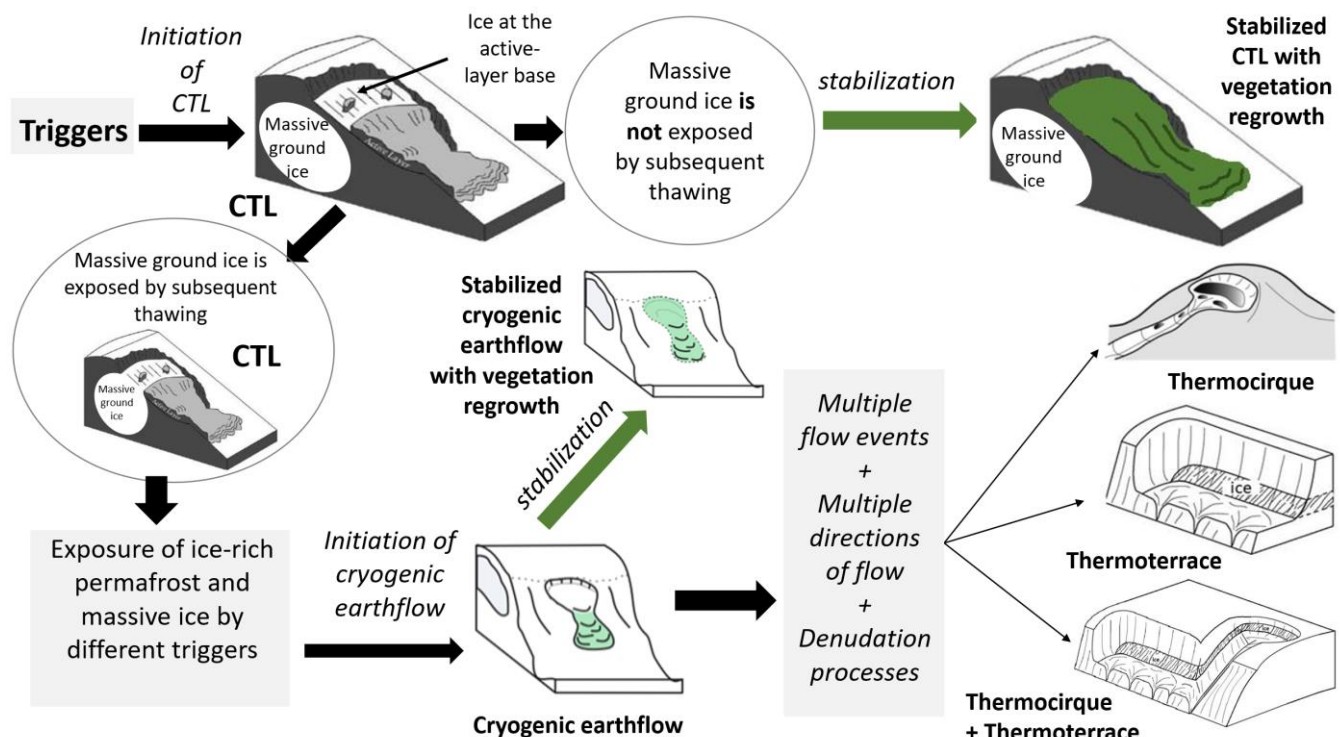

487

Figure 9 Conceptual diagram of the interrelation of different terms in the Russian literature used to describe the RTS formation
process and the resulting landforms. CTL stands for cryogenic translational landslide. Note that what is shown as massive ground
ice in the diagram may have different characteristics in different regions and could include buried glacial ice, thick ice layers, or
large syngenetic ice wedges.

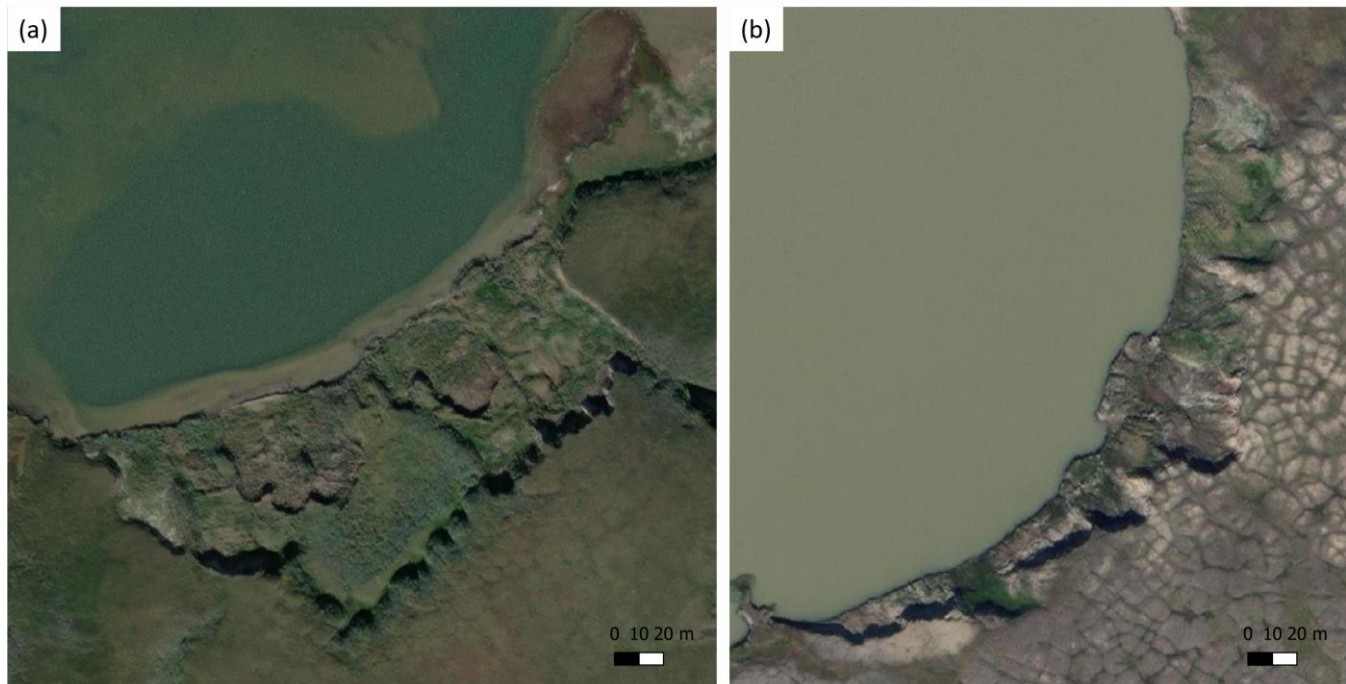

492

**Figure 10 Examples of combined RTS morphologies with thermoterraces and thermocirques. (a) Thermocirque(s) growing into a stabilized thermoterrace in Central Yamal, West Siberia, Russia as seen in a WorldView-3 satellite image from August 2018 (ESRI satellite basemap). (b) Thermocirque and thermoterrace merging at the coast into one outline in Western Yamal, West Siberia, Russia as seen in a WorldView-3 satellite image from August 2018 (ESRI satellite basemap). ESRI basemap has the following credits: Esri, DigitalGlobe, GeoEye, i-cubed, USDA FSA, USGS, AEX, Getmapping, Aerogrid, IGN, IGP, swisstopo, and the GIS User Community.**

### 4.2. Overlap in terminologies

The terms described above refer to similar phenomena of the RTS formation process and resulting landforms, leading to inevitable similarities and overlaps but also differences (Table 2).

*Cryogenic translational landslide* corresponds to shallow active layer detachment slide in North American literature that is triggered by high pore-water pressure and low effective strength (Lewkowicz, 2007). *Cryogenic earthflow* corresponds to a deep ALD in the North American literature and is the very early stage of RTS formation. *Thermocirque* and *thermoterrace* signify the mature stage of RTS of different morphology.

The process of RTS formation can generally be described as a mass-wasting (landsliding) process resulting from the melting of massive ground ice exposed due to various triggers. Regardless of which terminology is used (*thermodenudation* in Russian literature or *thermokarst* in North American literature), it can be seen as a sequence of physical events (Fig.11): trigger, massive ground ice exposure, ice ablation, thaw-related mass movement, and landform formation. The first two physical events (trigger + massive ground ice exposure) are usually considered the RTS initiation stage. Triggers of massive ice exposure lead to mass-wasting and RTS landform occurrence.





**Table 2 Correspondence of the process, landform, and terminology in the different approaches currently used**

| Main physical process | Process term | | Resulting landform | | | |
|---|---|---|---|---|---|---|
| | North American literature | Russian literature | North American literature | | Russian literature | |
| | | | term | comment | term | comment |
| Mass-wasting on seasonal ice at the base of the active layer (within the active layer) | *Thermokarst (in wide definition)* | *Thermodenudation (in narrow definition)* | Active layer detachment slide (ALD) | shallow, relatively dry | Cryogenic translational landslide | Can trigger massive ground ice exposure |
| Mass-wasting on massive ground ice (upper part of the permafrost) | | | | deep, saturated | Cryogenic earthflow | The initial stage of retrogressive thaw slump formation |
| Mass-wasting due to the exposure and further thawing of ice-rich permafrost or melting massive ground ice plus other denudational processes resulting in concave hollows | | | Retrogressive thaw slump (RTS) | - | Thermocirque | The mature stage of retrogressive thaw slump development. Morphology: mostly horseshoe shape |
| | | | | | Thermoterrace | The mature stage of retrogressive thaw slump development. This landform is initiated on coastal bluffs or large lakes with coastal erosion playing a role. Morphology: mostly elongated |


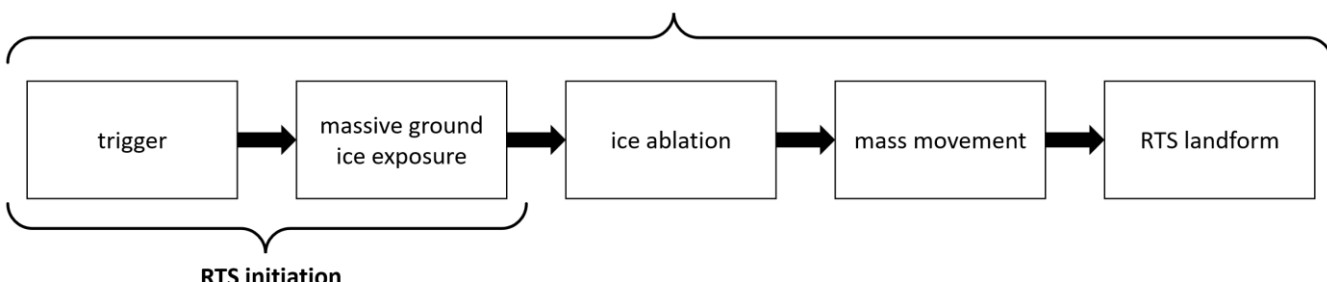


**Figure 11 Broadly accepted sequence of physical events of RTS formation.**

### 4.3. Limitations of divergent terminology

All the terms used to explain RTS formation both in North American and Russian literature have their limitations. The usage of a single term Thermokarst for a wide variety of processes leads to confusion about the direction of the physical process happening: whether it is the vertical lowering of the surface in the case of thermokarst lakes or lateral mass movement in the case of RTS occurrence. Since the term Thermokarst in this case incorporates both directions of the process, it is crucial to clearly state that RTS formation implies back-wearing thermokarst (or hillslope thermokarst). Another confusion can appear when talking about mass movements that are deeper than the active layer and slide on the surface of massive ground ice. In the North American literature, such landslides can still be called active layer detachment slides. However, since these mass movements expose massive ground ice, the retrogression can already start, which means that it is already an early-stage RTS. Such mass movements on the surface of massive ground ice are called cryogenic earthflows and are considered early-stage RTS in Russian literature. However, it is difficult to distinguish an early-stage RTS (cryogenic earthflow) from a mature-stage RTS (thermocirque) since mature RTS can also be of small sizes. Clear separation of these two categories is almost impossible with remote sensing data and is quite demanding in the field since it requires thorough knowledge of the environment and the dynamics of each RTS.

The definitions of *thermocirque* and *thermoterrace* present in the literature are based on the morphology of the features. Considering morphology as a distinguishing factor can be subjective since no established curvature values exist in the literature to differentiate them. In some cases, a thermoterrace can appear more curved, rather resembling a thermocirque. In contrast, a thermocirque can further elongate in width following the initial shape of massive ground ice (e.g., Fig.1 in Swanson and Nolan, 2018), while its mudflow can reach the neighboring water body base level.

### 4.4. RTS definition in the Glossary

With a large number of recent RTS mapping studies in different permafrost regions, it has become clear that RTS characteristics and morphologies vary widely, that RTS can occur in a range of different permafrost and ground ice settings, and feature processes important for understanding their dynamics and environmental impacts. However, these aspects are not

yet covered by the current definition of a "retrogressive thaw slump" in the International Permafrost Association Multi-
Language Glossary of Permafrost and Related Ground-Ice Terms (van Everdingen, 2005) (see Sect. 3.2.1). This definition is
rather short and describes a portion of RTS characteristics, it is limited in its scope and does not capture the full breadth of
RTS variability emerging from the many studies. In particular, the definition only focuses on the active stage of RTS, while
the polycyclic nature of many RTS also includes the stages of stabilization without activity. Moreover, this definition does not
reflect the variety of possible morphologies as horseshoe-like (thermocirques) or elongated along the coast (thermoterrace)
and different stages of the landform evolution. Furthermore, some other settings also feature slump-like landforms that exhibit
a similar headwall backwasting but were not covered in this review. Such slumps for example occur on recent dead-ice
moraines that experience retrogressive rotational sliding or back slumping of the ice-cored slopes (Kjær and Krüger, 2001).
Thus, a clear distinction should be drawn in the definition. We recommend considering these points when preparing the next
International Permafrost Association Multi-Language Glossary of Permafrost and Related Ground-Ice Terms.
**4.5. Missing terminology**
Our review of morphologic elements of RTS (see Sect. 3.1) showed that there is no term to describe unthawed permafrost
remnants within a slump floor. The term *slump block*, in our opinion, fits the best to explain pieces of soil with vegetation that
move downwards while the term *remnant island* sounds rather confusing because it does not assume the moving nature of
such a feature. We rather suggest using the term *remnant island* to describe unthawed permafrost remnants within a slump
floor. These remnant islands are generally larger than slump blocks and do not move since they still have unthawed cores. An
example of such a remnant island is shown in Fig.12.

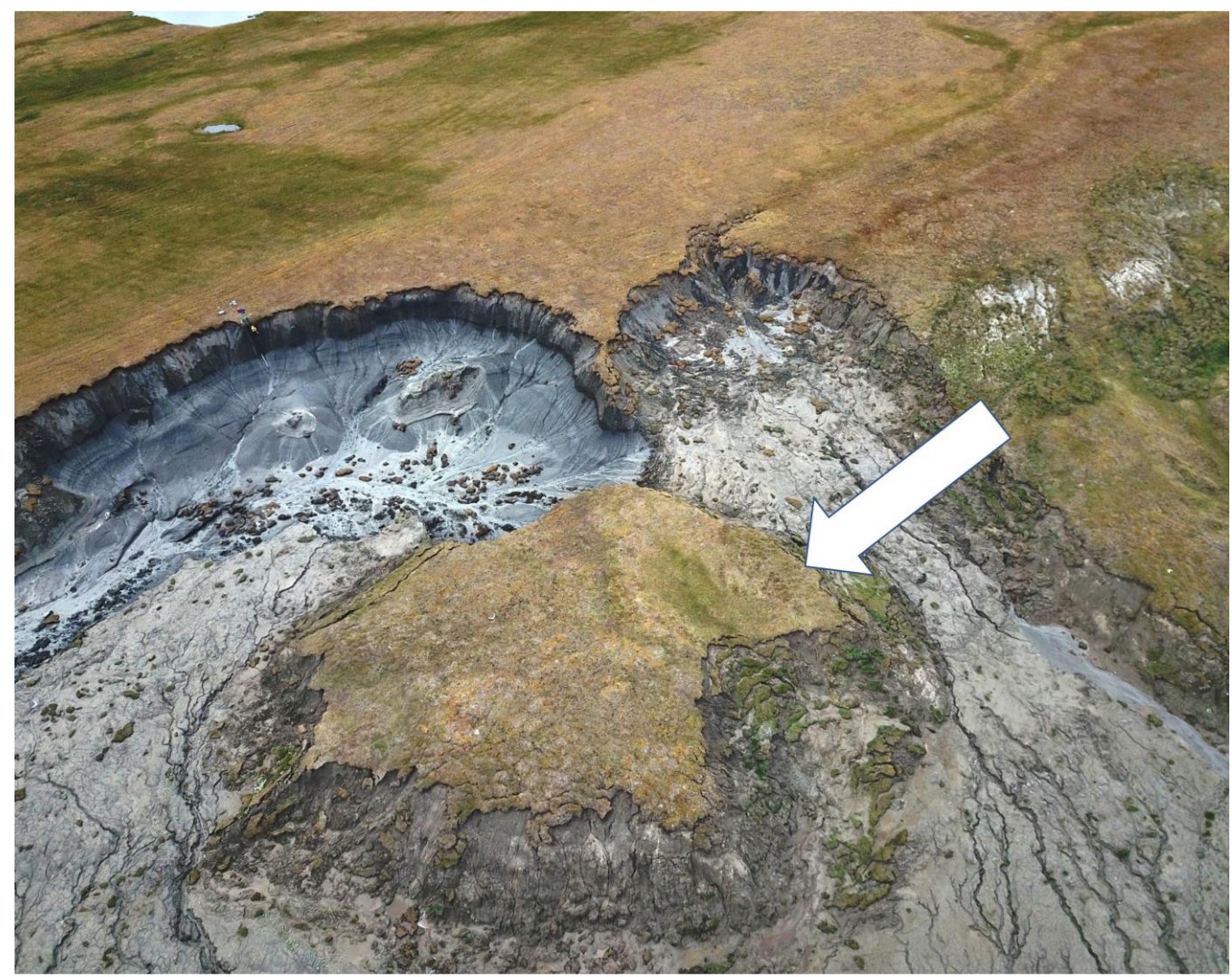

**Figure 12 Example of an unthawed remnant island (indicated with white arrow) within a slump floor of RTS in Yugorsky Peninsula,**
**European Arctic, Russia, September 2019. UAV photo: Nina Nesterova.**

## 5 Conclusions

Retrogressive thaw slumps are complex permafrost region landforms that despite recent wide scientific interest are still studied
very differently in terms of terminology. Based on our review of the literature and terminologies we draw the following
conclusions:
● The RTS formation process is currently explained with two different terms (thermokarst and thermodenudation) in
the North American and Russian literature based on different theoretical views that were formed in the 20th century.

- RTS is a general umbrella term applied to different stages of landform activity and also a variety of mass-wasting landforms on slopes with ice-rich permafrost (thermocirque/thermoterrace).
- RTSs can differ in shape, triggers, ground ice types, position in the relief, activity, concurrent processes, and spatial aggregation.
- For active RTS we identified four essential morphologic parts (headwall, slump floor, mudflow, edge), while nine additional parts may or may not be present in an RTS.

The study of RTS formation and accompanying processes is important to better understand how rapid mass wasting on permafrost slopes can mobilize sediment, meltwater, carbon, and nutrients, how biogeochemical dynamics are influenced by specific processes during the RTS formation and growth, and how RTS may pose hazards to infrastructure. More clarity on used terminology and scientific views will foster this understanding and can guide new research.

**Author contribution**

NN: Conceptualization, Resources (literature sources), Investigation, Writing – original draft preparation. ML: Conceptualization, Supervision, Writing – review & editing. AK: Supervision, Writing – review & editing. HL: Conceptualization, Supervision, Writing – review & editing. IT: Resources (literature sources), Writing – review & editing. IN: Writing – review & editing. AV: Writing – review & editing. GG: Conceptualization, Supervision, Writing – review & editing.

**Competing interests**

The authors declare that they have no conflict of interest.

**Acknowledgments**

We would like to acknowledge and appreciate the helpful discussion of RTS terminology with S. Kokelj and J. van der Sluijs during the early stage of the manuscript. This review is part of the efforts in the RTSInTrain Action Group led by A. Liljedahl on retrogressive thaw slumps by the International Permafrost Association. ML and IT were funded by the state assignment of the Ministry of Science and Higher Education of the Russian Federation (topic No. FWRZ-2021-0012). IN and GG were supported by the Google Permafrost Discovery Gateway. AK was funded by the state assignment «Evolution of the cryosphere under climate change and anthropogenic impact» (#121051100164-0). NN was funded by a DAAD fellowship (Grant #57588368).

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
