# Peer review of "Review article: Retrogressive thaw slump characteristics and terminology"

_EGUsphere, 2023_

## Author Comment (AC1)

**Author Response to Reviewer #1.**

*The comments by Reviewer #1 are in black. The author's responses are in blue. The changes suggested to the revised manuscript are in green.*

*Anonymous Referee #1*

*Referee comment on "Retrogressive thaw slump theory and terminology" by Nina Nesterova et al., EGUsphere [preprint], https://doi.org/10.5194/egusphere-2023-2914, 2024.*

Nesterova et al. present an overview of taxonomies to describe retrogressive thaw slumps, their morphological characteristics and associated geomorphic processes. To bridge the disparate terminologies, the authors present and contrast taxonomies from the Russian and Western literature.

I laud the overall goal and see this contribution as an important step toward reconciling the disparate schools. However, it is difficult to say to what extent the present manuscript achieves this goal. The manuscript could be strengthened by clear definitions for all the terms it introduces, by drawing a sharp boundary between definitions and observations, and by more precise language. Currently, there is a risk the article will only be of interest to a niche audience. Clear definitions and descriptions would strengthen the manuscript substantially, as they would enable researchers from diverse backgrounds to thoroughly appraise the existing literature. Because similar issues pervade periglacial science (e.g., patterned ground), it could serve as a role model for review papers on various types of landforms, processes, etc.

We would like to thank the reviewer for finding the time to review our manuscript. We highly appreciate valuable comments that help to improve the quality of the manuscript.

Our goal is to present a critical overview of the properties and terminology from the literature related to RTS phenomena. Since the recent attempt to bridge disparate terminologies was unsuccessful due to present disagreement within the research community, this manuscript aims to present a non-biased overview without expressing the authors' position. Moreover, we aim to submit the review to the Encyclopedia of Geosciences collection, where no-position is one of the main criteria: "A review paper is not a position paper. In the case of topics under dispute, a fair and balanced overview over the main positions is required."

We have reworded the aim in the Introduction to express the aim of a balanced and no-position literature review explicitly (particular changes in bold):

"This work aims to provide clarifications on the existing terminology of RTS phenomena and ease the understanding of published studies. The paper presents commonly observed RTS characteristics and a **neutral** review of existing RTS terminology in the literature. Our review considers a broad variety of RTSs in the Northern Hemisphere."

We fully agree on the need to draw a sharp boundary between definitions and observations to make the manuscript easier to follow for the readers. To address this issue, we have restructured the paper to separate "observed characteristics" and "terminology" as follows:

To ensure that definitions of various terms are easier to follow we have dedicated a separate subsection for each term we are describing.

1) Definitions

I encourage the authors to include clear definitions that enable a researcher with limited prior knowledge of these taxonomies to classify a given landform. If no prior or conflicting definitions are available, your guidance will be all the more valuable. Currently, almost none of the landforms are defined. I provide a few examples in the following.

To ensure that definitions of various terms are easier to follow we have dedicated a separate subsection for each term we are describing. We have reworded all the definitions based on the reviewed literature. Since the aim is to perform a no-position balanced review we avoided presenting our own definitions.

a) The Canadian RTS glossary entry is included here and criticized for, among other things, not including stabilized landforms. What would be a useful definition? What is the definition implicitly used in the remainder of the manuscript? Is an RTS a landform (as suggested by the glossary entry) or is RTS also a process (as mentioned in the conclusion, but barely developed in the main body of the document)?

Since we have restructured the paper, the definition of RTS in the International Permafrost Association Multi-Language Glossary of Permafrost and Related Ground-Ice Terms (van Everdingen, 2005) that we refer to is currently presented in section 3.2.1. Terminologies used in the literature → Landforms → Retrogressive thaw slump (RTS). This paragraph only states the current definition agreed upon within the International Permafrost Association:

"**3.2.1. Retrogressive thaw slump (RTS)**

According to the International Permafrost Association Multi-Language Glossary of Permafrost and Related Ground-Ice Terms (van Everdingen, 2005), RTS is defined as: "A slope failure resulting from thawing of ice-rich permafrost. Retrogressive thaw slumps consist of a steep headwall that retreats in a retrogressive fashion due to thawing and a debris flow formed by the mixture of thawed sediment and meltwater that slides down the face of the headwall and flows away. Such slumps are common in ice-rich glaciolacustrine sediments and fine-grained diamictons.""

We have moved the critical review on this definition to Discussion: 4.4. RTS definition in the Glossary. To maintain a neutral and balanced review, we have deliberately refrained from providing our own definitions. Thus, we only provided recommendations for the future authors preparing the International Permafrost Association Multi-Language Glossary of Permafrost and Related Ground-Ice Terms:

"**4.4. RTS definition in the Glossary**

With a large number of recent RTS mapping studies in different permafrost regions, it has become clear that RTS characteristics and morphologies vary widely, that RTS can occur in a range of different permafrost and ground ice settings, and feature processes important for understanding their dynamics and environmental impacts. However, these aspects are not yet covered by the current definition of a "retrogressive thaw slump" in the International Permafrost Association Multi-Language Glossary of Permafrost and Related Ground-Ice Terms (van Everdingen, 2005) (see Sect. 3.2.1). This definition is rather short and describes a

portion of RTS characteristics, it is limited in its scope and does not capture the full breadth of RTS variability emerging from the many studies. In particular, the definition only focuses on the active stage of RTS, while the polycyclic nature of many RTS also includes the stages of stabilization without activity. Moreover, this definition does not reflect the variety of possible morphologies as horseshoe-like (thermocirques) or elongated along the coast (thermoterrace) and different stages of the landform evolution. Furthermore, some other settings also feature slump-like landforms that exhibit a similar headwall backwasting but were not covered in this review. Such slumps for example occur on recent dead-ice moraines that experience retrogressive rotational sliding or back slumping of the ice-cored slopes (Kjær and Krüger, 2001). Thus, a clear distinction should be drawn in the definition. We recommend considering these points when preparing the next International Permafrost Association Multi-Language Glossary of Permafrost and Related Ground-Ice Terms.''

b) Shallow landslides: No definition of a "cryogenic translational landslide" is provided. Do these have to be translational (as the name suggests), by definition? Is the triggering by high pore-water pressure required by definition, or is this commonly observed or inferred for landforms that fall within the definition? For the ice whose melt induces pressurization: Does it have to be seasonal (and how can you tell, i.e., is this a useful definition) and does it have to be at the base of the active layer. A clear definition would help me determine whether detachments of the organic layer in discontinuous permafrost are CTLs, or shallow landslides on slopes underlain by taliks. The same concerns apply to cryogenic earthflows.

Thank you for emphasizing the importance of rewording the definitions clearly. We have summarized the definition of the cryogenic translational landslides and cryogenic earthflows mentioned in several publications in the literature:

"**3.2.2. Cryogenic earthflow**

Here, it is worth defining cryogenesis as a set of thermophysical, physicochemical, and physicomechanical processes occurring in freezing, frozen, and thawing deposits (van Everdingen, 2005). The word cryogenic is usually used to describe the periglacial nature of the processes.

The term cryogenic earthflow was introduced by Leibman (1997, in Russian) meaning a viscous or viscoelastic flow of water-saturated soil of the active layer sliding on the surface of massive ground ice bodies or the table of ice-rich permafrost. The examples of cryogenic earthflows in Central Yamal are demonstrated in Fig.4.

<...>

**3.2.6. Cryogenic translational landslide**

The term cryogenic translational landslide (CTL) was suggested by Kaplina (1965, in Russian), and the definition was later elaborated in further publications based on observations in Central Yamal, Russia (Leibman and Egorov, 1996; Leibman, 1997; Leibman et al., 2014). The definition of CTL summarized from the abovementioned publications can be phrased as single-time lateral displacement of thawed soil block sliding on the surface of the seasonal ice formed at the active layer base. This type of seasonal ice is formed due to the active layer's upward freezing, ice aggradation at the base of the active layer, and later melting (Leibman et al., 2014; Lewkowicz, 1990). Examples of CTL in Central Yamal are shown in Fig.7.''

c) Thermocirques and thermoterraces: The paragraph starting at line 421 seems to assume the reader knows what is being referred to. In general, the distinction appears to be based on genesis rather than morphology, but it is not clear to me to what extent they are to be discriminated based on the morphology. For instance, Fig. 7b shows a thermocirque along a lake. Where did it initiate, and unless precise information is available, how was its present-day morphology taken into consideration to classify it as a thermocirque? If the location of initiation is the determining factor, a length scale could be informative: e.g., <=3 vs >3 m from the waterline at the time of initiation (averaged over at least 1 day).2) Description vs. definition

We have rewritten all the definitions based on how these features were defined or described in the literature. The definitions of *Thermocique* and *Thermoterrace* are currently worded as follows:

"**3.2.3. Thermocirque**

The term thermocirque was first mentioned by Czudek and Demek (1970, in English) to describe "amphitheatrical hollows" that occur after ice wedge melt in the gullies at the river banks in Yakutia (Russia). Thermocirques according to the authors had "a vertical and overhanging slope at the head and an uneven floor". In Russian-language literature, the term thermocirque was sometimes called by interchangeable term "thermokar" when describing a round or cirque-like hollow at the river banks or the lake shores composed of icy permafrost (Grigoriev and Karpov, 1982, in Russian; Voskresenskii, 2001, in Russian). Following the development of theoretical concepts of cryogenic landsliding (Sect. 3.2.3 and 3.2.4) the term thermocirque was defined as an extensive landform resulting from a series of multi-aged cryogenic earthflows (Leibman, 2005, in Russian; Leibman et al., 2014, in English). The scheme visualizing thermocirque formation and the example of the thermocirque in Central Yamal, Russia are demonstrated in Fig.5.

**3.2.4. Thermoterrace**

The term thermoterrace was first mentioned by Ermolaev (1932, in Russian) to describe "picturesque outcrops of ice falling vertically onto a narrow, 1-2 m wide space located along the seashore along the edge of the ice wall that can reach 30-35 m". The local term to describe these icy cliffs was muus kygams - muus кьham in Yakutian language (Ermolaev, 1932). The more precise definition of thermoterrace was given by Zenkovich and Popov (1980) as a terrace-like area in the upper part of the icy cliff at the seashore that results from the cliff retreat due to the thermal influence of warm air and solar radiation. Thermoterraces were reported to reach up to a few km in length along the coast and more than 200 m in width (Are et al., 2005). A scheme visualizing thermoterrace formation based on Kizyakov (2005) and an example of a thermoterrace on the Bykovsky Peninsula, Yakutia, Russia are shown in Fig.6."

Thank you for pointing out the confusion with the distinguishing factor for definitions of thermocirques and thermoterrace. Unfortunately, the definitions in the literature do not provide a clear answer as to whether it is strictly genesis-based or morphology-based. To elaborate on this issue, we have enlarged the paragraph in section *4.3 Limitations of divergent terminologies* providing the reference to the paper where this problem was already discussed:

"The definitions of *thermocirque* and *thermoterrace* present in the literature do not draw an explicit distinction between these two landforms leading to the confusion of whether the discrimination is based on morphology or the genesis. On the one hand, considering the morphology as the only distinguishing factor can be misleading since in some cases, a thermoterrace can appear more curved, rather resembling a thermocirque. In contrast, a thermocirque can further elongate in width following the initial shape of massive ground ice (e.g., Fig.1 in Swanson and Nolan, 2018), while its mudflow can reach the neighboring water body base level. On the other hand, a distinction based only on genesis has never been mentioned in the literature in the context of the terms thermocirque and thermoterrace. Moreover, separating the landforms based on their genesis requires retrospective analysis of RTS formation and thus multi-temporal observations and data of an RTS. In a recent publication, Leibman et al. (2023) opted not to use the terms *thermocirque* and *thermoterraces* for different genesis but only for different morphologies. Instead, a clear genesis-based distinction was established by proposing the terms *open cryogenic-landslide landform* for RTSs formed by the retreat of the coastal bluff influenced by wave action, and *closed cryogenic-landslide landform* for RTSs formed inland."

I was struggling to distinguish which statements were definitional vs. descriptive. This relates partially to the lack of clear definitions (see 1), but also to ambiguous language. These ambiguities further made it impossible for me to identify the theoretical foundations alluded to in the title. I would expect any scientific theory to make testable predictions, based on a coherent set of clearly defined processes/quantities/observables.

We completely agree on the clutter caused by mixing definitions and descriptions. To address this issue we, as stated above, restructured the paper and provided clear and concise definitions in the "Terminologies used in the literature" section.

We fully understand the confusion regarding "theory" in the title, thus we have adjusted the title to make it reflect the content of the manuscript: **"Review article: Retrogressive thaw slump characteristics and terminology"**.

Several examples of ambiguous language are provided below.

- The authors note there is little evidence that "aspect defines RTS occurrence." I suspect this is a descriptive statement, meaning that the observed regional associations between aspect and slump occurrence are variable.

We have removed this deterministic sentence, leaving only regional findings on the presence or the absence of the correlation:

"RTSs occur on a great variety of slope aspects. While some studies investigating different regions across the Arctic reported that their observed RTSs tended to have different prevailing slope orientations (Kokelj et al., 2009; Lacelle et al., 2015; Jones et al., 2019; Nesterova et al., 2021; Bernhard et al., 2022), several other studies found that higher RTS ablation rates and headwall retreat (see Sect. 3.1.1) are related to southern aspects (Lewkowicz, 1987a; Grom and Pollard, 2008; Lacelle et al., 2015). However, several other studies did not find any link

between the slope aspect and RTS activity (Wang et al., 2009; Nesterova et al., 2021; Bernhard et al., 2022). Bernhard et al. (2022) suggested that differences in the RTS aspect may be explained by regional geological history that defines ice content and ice distribution, which are the main factors of RTS occurrence (Mackay, 1966; Kerfoot, 1969)."

- On line 74, it is stated that slumps "develop in a polycyclic fashion". This statement is presented as a universally valid declaration. Does slumping have to be polycyclic?

Thank you for this comment! We fully agree that not all RTSs exhibit polycyclic behavior. Thus, we reworded the sentence:

"RTSs **can** develop in a polycyclic fashion, which means they can be active, then temporarily stabilize, and also reactivate again (Mackay, 1966; Kerfoot, 1969; Kokelj et al., 2009). Yet some may end off in one cycle."

3) Precise language advised

I think the manuscript would benefit from more precise language in many places. Vague statements are difficult to falsify.

Consider specifying the spatial and temporal scales in the descriptions. For instance, in line 395, CTLs are described as rapid. How rapid? Elsewhere, they are described as "very dynamic". What does this mean?

Thank you for pointing this out. Since we have not found quantitative estimations of the mass movement speed, we removed the attribute "rapid" in the text:

"**3.2.6. Cryogenic translational landslide**

The term cryogenic translational landslide (CTL) was suggested by Kaplina (1965, in Russian), and the definition was later elaborated in further publications based on observations in Central Yamal, Russia (Leibman and Egorov, 1996; Leibman, 1997; Leibman et al., 2014). The definition of CTL summarized from the abovementioned publications can be phrased as single-time lateral displacement of thawed soil block sliding on the surface of the seasonal ice formed at the active layer base. This type of seasonal ice is formed due to the active layer's upward freezing, ice aggradation at the base of the active layer, and later melting (Leibman et al., 2014; Lewkowicz, 1990). Examples of CTL in Central Yamal are shown in Fig.4."

It is claimed that the "spatial distribution of ground ice determines the spatial extent of RTS." This is a strong deterministic statement, but the subsequent paragraph does not provide

quantitative information. Do the climatic conditions play any role, or sediment properties? What are the relevant spatial and temporal scales?

We have softened this statement:

"**2.3. Ground ice**

**A high excess ground ice content is a prerequisite for RTS occurrence.** The shallower the ground ice table the higher the likelihood that seasonal thawing will reach and start melting the ice, potentially triggering the initiation of the RTS. Regions with abundant ground ice presence in Canada feature widespread and ubiquitous slumps (Lamothe and St-Onge, 1961; Mackay, 1966; Kokelj et al., 2017). Similar observations were reported for Central Yamal, Russia (Babkina et al., 2019). RTS in areas with a thinner ground ice-rich layer tend to stabilize faster due to the rapid ice exhaustion (Kizyakov, 2005). The type of ground ice and its local distribution can define some morphologic characteristics of RTS (see Section 3.1) and affect retreat rates. For example, RTS forming in syngenetic ice-rich Yedoma deposits with polygonal ice wedges are usually accompanied by the presence of baydzherakhs (conical remnant mounds, for details, see Section 3.1.6) on the slump floors. De Krom and Pollard (1989) found that on Herschel Island, Canada, large ice wedges melted more slowly than the enclosing massive ground ice. **While abundant ground ice is necessary for RTS formation it is not the only control for RTS occurrence.**"

It is claimed that "ablation happens only in summer when the air temperature is above 0C". Can it happen in the fall? Can it happen under strong radiation (e.g., Tibetan Plateau) when the 2m air temperature is <0C? See e.g. Lewkowicz 87.

We have removed this sentence.

4) Scope

a) Paraglacial phenomena

Slumps on moraines or debris-covered glaciers were not considered in the manuscript, but they were not explicitly ruled out either.

We mentioned dead ice backslumps in the Discussion under "4.4. RTS definition in the Glossary":

"**4.4. RTS definition in the Glossary**

With a large number of recent RTS mapping studies in different permafrost regions, it has become clear that RTS characteristics and morphologies vary widely, that RTS can occur in a

range of different permafrost and ground ice settings, and feature processes important for understanding their dynamics and environmental impacts. However, these aspects are not yet covered by the current definition of a "retrogressive thaw slump" in the International Permafrost Association Multi-Language Glossary of Permafrost and Related Ground-Ice Terms (van Everdingen, 2005) (see Sect. 3.2.1). This definition is rather short and describes a portion of RTS characteristics, it is limited in its scope and does not capture the full breadth of RTS variability emerging from the many studies. In particular, the definition only focuses on the active stage of RTS, while the polycyclic nature of many RTS also includes the stages of stabilization without activity. Moreover, this definition does not reflect the variety of possible morphologies as horseshoe-like (thermocirques) or elongated along the coast (thermoterrace) and different stages of the landform evolution. **Furthermore, some other settings also feature slump-like landforms that exhibit a similar headwall backwasting but were not covered in this review. Such slumps for example occur on recent dead-ice moraines that experience retrogressive rotational sliding or back slumping of the ice-cored slopes (Kjær and Krüger, 2001).** Thus, a clear distinction should be drawn in the definition. We recommend considering these points when preparing the next International Permafrost Association Multi-Language Glossary of Permafrost and Related Ground-Ice Terms."

b) Stabilization

A binary distinction between active thaw slumps and stabilized thaw slumps is made, with active thaw slumps featuring exposed ice (e.g., 3.5.1). Conversely, Kokelj et al. 2015 and Zwieback et al. 2020, amongst others, described thaw slumps that remained active on ~annual time scales despite featuring intermittent or even a persistent sediment cover. Would a more nuanced view on sediment cover and stabilization strengthen the manuscript?

Thank you, a lot, for pointing this out! This is an important note and we elaborated on this in the text (particular changes in bold):

"**2.5. Polycyclicity**

RTSs can develop in a polycyclic fashion, which means they can be active, then temporarily stabilize, and also reactivate again  (Mackay, 1966; Kerfoot, 1969; Kokelj et al., 2009). Yet some may end off in one cycle. RTSs can be considered active when there is an ongoing ablation of the exposed ice and thawed material is transferred downslope. **Some studies reported continued headwall retreat and thawed sediment fluxes even in slumps where the ice was covered by the sediments (Kokelj et al., 2015; Zwieback et al., 2020). The reasons for these sediment-covered slumps to retain activity were heavy rainfalls and unsuppressed heat flux to the ice.**

RTSs can stabilize mostly for two reasons: 1) exposed ground ice has completely melted, or 2) the exposed ice is re-buried by sediments and thermally fully insulated from further melting (Burn and Friele, 1989). Once an RTS is stabilized, pioneer vegetation starts to grow in the slump floor. Vegetation in stabilized RTS can go through several stages of succession and for stabilized RTS in Yukon Territory, Canada, it was reported that forest and tundra communities were re-established after 35-50 years (Burn and Friele, 1989). Some researchers found that RTSs can be stabilized for up to several hundred years in West Siberia, Russia,

(Leibman and Kizyakov, 2007). Such long-term stabilized RTS are labeled in some studies as ancient (Nesterova et al., 2023).

New active RTS can form within the outline of another stabilized RTS, moreover, neighboring RTSs can grow and coalesce at some point (Lantuit and Pollard, 2008). This leads to the very complex spatial aggregation of nested and amalgamated RTSs of sometimes different ages. It raises additional challenges when delineating and mapping RTS and their characteristics (van der Sluijs et al., 2023; Leibman et al., 2023)."

c) Subjacent taliks and bay formation

The manuscript briefly mentions Kokelj et al. 2005, without describing the mechanisms involved. Also consider highlighting consequences of subsidence in the slump floor and below the adjacent waterbody, such as bay formation.

We have mentioned the article "The influence of thermokarst disturbance on the water quality of small upland lakes, Mackenzie Delta region, Northwest Territories, Canada" by Kokelj et al. (2005) in the Introduction as one of the examples of the RTS impact on the environment. In general, RTS influence on the environment including the consequent landform or bay formation deserves a separate literature review that will require a significant amount of time. Unfortunately, the next evolutionary step of RTS occurrence is out of the scope of the presented manuscript.

Minor points

l 166: It may be useful to consider differences between regions and landforms. For instance, many slumps on Banks Island feature a break in slope in the headwall, while many in the Anderson Plain/Tuktoyaktuk Coastlands do not.

We have highlighted in the text of the Introduction the possible regional diversities of RTSs in morphology and other characteristics. Moreover, we have added a figure with photos of RTSs in different regions of the Northern Hemisphere. The overview of regional differences of various RTS landforms is outside of the scope of this paper. It is a very interesting but time-consuming idea that can be implemented in a separate project with a significant amount of time scheduled to reach this goal.

"Figure 1 shows examples of different RTSs photographed across the Northern Hemisphere. RTSs exhibit regional variations in their appearance and characteristics."

l 230: What is a "cliff retreatment"? What do you mean by lower and upper edge?

Thank you for pointing out this typo with "cliff retreat", we have corrected it.

The terms lower and upper edge are used by some authors, i.e. Leibman et al. (2021). Fig 1 in this mentioned paper visualizes these morphological parts in a scheme. We have reworded the sentences:

"Furthermore, the edge of RTS is also sometimes classified into upper edge meaning the boundary line of active retreat of the headwall (Kizyakov et al., 2023), and lower edge meaning the boundary line of the cliff retreat for RTSs on the sea coasts (Leibman et al., 2021). **The face (cliff) from the lower edge of coastal RTS to the beach level has been called a dropwall (Leibman et al., 2021) to differentiate this morphologic part of the RTS from the rest of the coastal cliff.**"

l 271: isolation->insulation

Thank you! Corrected.

l 408: Soils often exhibit plastic or pseudoplastic behavior

We have used "viscous and viscoelastic flow" as it is written by Leibman et al., (2014).

---

## Author Comment (AC2)

**Author Response to Reviewer #2.**

*The comments by Reviewer #2 are in black. The author's responses are in blue. The changes suggested to the revised manuscript are in green.*

*Anonymous Referee #2*

*Referee comment on "Retrogressive thaw slump theory and terminology" by Nina Nesterova et al., EGUsphere [preprint], https://doi.org/10.5194/egusphere-2023-2914, 2024.*

As review 2, who was asked and accepted late, I both have read the manuscript and review 1. In my review I try not to repeat many of the comments from review 1, which all are valid, and I totally agree with those statements.

I was very interested in the title and the importance of RTS in a time of permafrost degradation and thaw, making these landforms a very visible witness of climate change. While acknowledging the attempt to review these features, I struggle with the paper outline and writing. The following issues arise, partly also mentioned by reviewer 1:

We would like to express our gratitude for taking the time to review our manuscript and providing feedback and suggestions to improve its quality! We have worked on rewriting the paper to address the main issue of the clarity and understandability of the manuscript.

1. The paper first introduces RTS incl. history (chapter 1), then it defines RTS (chapter 2) and describes common morphological features (chapter 3), while the discuss two divergent views of RTS, starting again with an historical background (chapter 4). This is confusing, and should be changed before publication, and I do not follow the motivation to structure the paper like that. I would recommend moving parts of the "historical background" into the start of the review, maybe into the Introduction. For a review paper this is an interesting knowledge to start with.

We fully agree that the current structure may appear confusing. To address this issue, we have restructured the paper in a way that should be easier to follow:

1 Introduction
2 Observed characteristics of retrogressive thaw slumps
      2.1. Morphometry and dynamics
      2.2. Position and topography
      2.3. Ground ice
      2.4. Triggers
      2.5. Polycyclicity
      2.6. Concurrent processes

3 Terminologies used in the literature
       3.1. Morphologic parts
              3.1.1. Headwall and Side-walls
              3.1.2.  Slump floor or Scar
              3.1.3. Mudpool and Mudflows
              3.1.4. Mud gullies and levees
              3.1.5. Slump block
              3.1.6. Baydzherakh(s)
              3.1.7. Evacuation channel
              3.1.8. Debris tongue
              3.1.9. Edge and dropwall
       3.2. Landforms
              3.2.1. Retrogressive thaw slump (RTS)
              3.2.2. Cryogenic earthflow
              3.2.3. Thermocirque
              3.2.4. Thermoterrace
              3.2.5. Active layer detachment slide
              3.2.6. Cryogenic translational landslide
       3.3. Formation process
              3.3.1. Thermokarst
              3.3.2. Thermodenudation
4 Discussion
       4.1 Divergent terminologies
       4.2. Overlap in terminologies
       4.3. Limitations of divergent terminologies
       4.4. RTS definition in the Glossary
       4.5. Missing terminology
5 Conclusions

We have moved the part about the historical roots of the terms (previously called "Historical background") to the Discussion under 4 Discussion → 4.1 Divergent terminologies, where we explain in detail the origin of existing disparate terms. Thus, the figure "The chronology of the usage of different terms by selected most cited authors in the 20th century…" is also moved there. Moreover, we enlarged the Introduction, including some additional historical background (particular changes in bold):

"<…>**Historically, RTS research started with the first mention of exposed ice in a retrogressive thaw slump probably dates back to 1881 by Dall in his publication on observations in Alaska (Dall, 1881) The first intensive studies on RTSs were conducted much later in the latter half of the 20th century in Canada (Lamothe and St-Onge, 1961; Mackay, 1966; Kerfoot, 1969) and Siberia (Popov et al., 1966; Czudek and Demek, 1970). These studies on RTSs were field-based and focused on ground ice, morphometry, and dynamics. The publications were written either in English or Russian language with different terms applied to these landforms depending on scientific approaches. Unfortunately, the level of knowledge exchange and reciprocal citation among RTS researchers from Canada and the USSR was relatively low, leading to the establishment of disparate views and terminology for RTS used in the literature.**

The strong rise in scientific exchange and international collaborations at the end of the 20th century, including joint expeditions within the permafrost community in general and within the topic of RTS in particular (i.e., Vaikmäe et al., 1993; Ingólfsson, and Lokrantz, 2003; Are

et al., 2005), as well as the emergence of remote sensing methods substantially broadened the scope of RTS research (Romanenko, 1998; Lantuit and Pollard, 2005; Lantz and Kokelj, 2008; Leibman et al., 2021). Today, a large body of recent literature predominantly focuses on monitoring RTS activity by measuring retreat rates (Kizyakov et al., 2006; Wang et al., 2009; Laccelle et al., 2010) and volume changes (Kizyakov et al., 2006; Clark et al., 2021; Jiao et al., 2022; Bernhard et al., 2022), identifying driving factors (Harris and Lewkowicz, 2000; Lacelle et al., 2010), or more generally mapping of RTSs (Pollard, 2000; Lipovsky and Huscroft, 2006; Khomutov and Leibman, 2008; Swanson, 2012; Segal et al., 2016). Recent publications on RTS mapping notably shifted away from a focus on geological and geomorphological aspects to developing advanced methodologies of RTS detection and classification using spatially and/or temporally high-resolution remote sensing data and digital elevation data, frequently employing artificial intelligence methods (Huang et al., 2020; Nitze et al., 2021; Yang et al., 2023).<…>"

2. The authors should review the common knowledge and discuss divergent views in a discussion chapter (which now is short and not really a discussion) or focus the paper on the different views in Russian and American literature as an example of divergent views, and come with recommendation on a common strategy. Now, the study is neither of those two.

Thank you for pointing this out. Since we aimed to review the observed characteristics of RTSs and the terminology used in the literature, we restructured the paper the way that Section 2 "*Observed characteristics of retrogressive thaw slumps*" presents the observed and described properties of RTS mentioned in the literature. Section 3 "*Terminologies used in the literature*" presents the terms (and their definitions) used in the literature to describe the naming of "*3.1. Morphologic parts*", "*3.2. Landforms*" and "*3.3. Formation process*". The Discussion Section presents an in-depth discussion on the origin and some particularities of "*4.1 Divergent terminologies*", also "*4.2 Overlap in terminologies*" and "*4.3 Limitations of divergent terminologies*". The Discussion also consists of the recommendations for the future definition of the RTS term in the next IPA Glossary ("*4.4 RTS definition in the Glossary*") and suggested term for the feature that missed the naming in the literature ("*4.5 Missing terminology*").

3. Because of that the paper is very hard to follow, the start of the manuscript is chopped in few descriptive chapters of landform details without illustration (move Fig. 1), incl a large table (maybe better off in an appendix). The second part is interesting incl. figure 3 is kind of illustrative, but is bot clearly connected to the first part.

We hope that restructuring the paper in the way described above will enhance the clarity and readability of the paper which consists of two separate parts: descriptive (observations) and definitions (terminology) parts followed by the discussion about terminology. Moreover, we

have added a figure with photos of RTSs in different regions of the Northern Hemisphere to the Introduction part for a better visual understanding of the described phenomena and their variability.

4. Concerning the discussion around landform and process, it reminds me a bit around discussion related to other landforms, such as rock glaciers, which is not always fruitful. In my understanding is RTS as term is similar to e.g. debris flow, this means a landslide process resulting in a landforms, which shape differs related to setting geological material the process is happening.

We thank the reviewer for this comment. We find the need for a critical unbiased review of the existing terminology related to RTS phenomena to avoid misunderstanding and misinterpretation of the landforms, features, and direction of the process. We have elaborated on the importance of the clarifications and discussion as well as the practical implementations of different terminology in the text of the Introduction (particular changes in bold):

"However, despite the increasing number of studies and strongly rising interest in RTS among the permafrost and remote sensing research communities, there is still no commonly agreed terminology on the RTS phenomenon. Various authors apply different terminology to describe the same morphology and processes or use the same terms for different processes. **This leads to several difficulties in communication about RTS within and across research communities. First of all, since the terminology is not always clearly defined or translated in the literature it can lead to potential misunderstandings about what exact features or processes have been investigated in a particular study. The confusion about the object of the study may cause incomparability of the datasets from different RTS studies. Furthermore, different labeling of the same features may result in a completely different image of the phenomena. For example, Nitze et al. (2024, in review) conducted an experiment where 12 domain experts from different countries manually mapped RTSs in Canada and Russia. The results demonstrated a large mismatch of the RTS labeling in Yakutia, Russia, which can be partially explained by different terminology used in the publications describing this region. The confusion in the terminology and labeling of RTSs can also affect the related studies on how RTSs impact hydrology, geochemistry, and ecology or their physical modeling, which is based on the established terms and concepts in the literature.**

This work aims to provide clarifications on the existing terminology of RTS phenomena and ease the understanding of published studies. The paper presents commonly observed RTS characteristics and a neutral review of existing RTS terminology in the literature. Our review considers a broad variety of RTSs in the Northern Hemisphere."

Do a thorough check of the references, e.g. Yershov (1998) in line 308 is not in the reference list. But I did not check everything here.

Thank you for noticing this issue! We have performed a thorough check and added 3 references that we forgot to put in the list and corrected the years in the other 3 references.

Precise language is important in review papers, as also review 1 mentioned. E.g. l. 135 makes no sense if the list all aspect instead of writing that "there is no preferred slope orientation". Also check definitions, e.g. you use the for me unknown term "baydzheraks" in l. 151 before you define it in chapter 3.5.6.

We fully agree with the importance of the precise language. To address this issue, we have reworded several statements as requested by Reviewer 1 and added the definitions in the first place, for example:

"For example, RTS forming in syngenetic ice-rich Yedoma deposits with polygonal ice wedges are usually accompanied by the presence of **baydzherakhs** (conical remnant mounds, for details, see Sect. 3.1.6) on the slump floors."

"•      the growth of a **debris tongue** (thawed sediments in a shape of a tongue, for details, see Sect. 3.1.8) can eventually obstruct a stream valley and lead to the increase of stream base level and further thermoerosion that can erode and expose the ground ice (Kokelj et al., 2015)."

We have omitted the list of slope aspects:

"**RTSs occur on a great variety of slope aspects.** While some studies investigating different regions across the Arctic reported that their observed RTSs tended to have different prevailing slope orientations (Kokelj et al., 2009; Lacelle et al., 2015; Jones et al., 2019; Nesterova et al., 2021; Bernhard et al., 2022), several other studies found that higher RTS ablation rates and headwall retreat (see Sect. 3.1.1) are related to southern aspects (Lewkowicz, 1987a; Grom and Pollard, 2008; Lacelle et al., 2015). However, several other studies did not find any link between the slope aspect and RTS activity (Wang et al., 2009; Nesterova et al., 2021; Bernhard et al., 2022). Bernhard et al. (2022) suggested that differences in the RTS aspect may be explained by regional geological history that defines ice content and ice distribution, which are the main factors of RTS occurrence (Mackay, 1966; Kerfoot, 1969)."

I really recommend a manuscript like this, and if thoroughly revised I am confident it will be read, commented and cited.

We would like to thank the reviewer once again for the valuable comments aimed at strengthening our manuscript!

---

## Author Response (AR3)

**Author Response to the second report of the Reviewer #1.**

*The comments by Reviewer #1 are in black. The author's responses are in blue. The changes suggested to the revised manuscript are in green.*

*Anonymous Referee #1. The second report*

*Referee comment on "Retrogressive thaw slump theory and terminology" by Nina Nesterova et al., EGUsphere [preprint], https://doi.org/10.5194/egusphere-2023-2914, 2024.*

Nesterova et al. have made numerous revisions to the initial submission. I laud the improvements to the clarity and completeness. I particularly appreciate the clearer distinction between definition and observation, and the more comprehensive scrutiny of the terminology.

We would like to thank the reviewer for taking the time to review our manuscript a second time. The attention to detail in the review has helped us enhance the clarity and readability of the text!

Two minor comments:

- Cryogenesis. I do not think the following sentence constitute a definition even in the loosest meaning of definition. By "... defining cryogenesis as a set of thermophysical, physicochemical, and physicomechanical processes occurring in freezing, frozen, and thawing deposits [,]" the authors do not identify what set is being referred to. Consider adopting descriptive language instead: "In the Russian literature, the term cryogenic is employed to refer to thermophysical, physicochemical, and physicomechanical processes occurring in freezing, frozen, and thawing deposits." Consider a similar approach for other vague definitions.

We agree that the definition of cryogenesis from the Glossary may appear unclear since this term is not that common in the English-language literature. We have elaborated in the text that the term is mostly used in Russian-language literature and omitted in English:

**"3.2.2. Cryogenic earthflow**

In Russian literature, the word *cryogenic* is usually used to describe the periglacial nature of the processes. It refers to thermophysical, physicochemical, and physicomechanical processes occurring in freezing, frozen, and thawing deposits (van Everdingen, 2005). This term is usually omitted in the literature in English (Poppe and Brown, 1976)."

For the remaining definitions, we either provided a direct citation or rephrased the definitions from the literature. For instance, the International Permafrost Association Multi-Language Glossary of Permafrost and Related Ground-Ice Terms (van Everdingen, 2005) was frequently cited with direct quotes for definitions of terms such as *retrogressive thaw slump* and *active layer detachment slide*. The definition of *thermodenudation* by Panov (1936) is also quoted, as it is a direct translation from the original source. We aimed to present the definitions as clearly as possible while preserving the original meaning from the literature.

- Thermokarst. Is French representative of the North American literature? The focus on water seems unusual, whereas for instance Kokelj and Jorgenson (Advances in Thermokarst

40 Research) or Farquharson et al. (Spatial distribution of thermokarst terrain in Arctic Alaska)
41 emphasize ground ice.

42 We agree that the definition of thermokarst by French (2018) presented in our manuscript
43 may not accurately reflect the current usage of the term in the context of RTS formation.
44 Therefore, we have replaced it with the definition provided by Kokelj and Jorgenson (2013):

45 "**3.3.1. Thermokarst**

46 The term thermokarst was first suggested by Ermolaev (1932) to describe the surface
47 subsidence due to the melting of ground ice as a similarity to the karst process by dissolution.
48 However, in the context of RTS formation processes the term thermokarst is mostly referred
49 to in the North American literature as a set of processes that lead to the occurrence of specific
50 landforms due to the thawing of ice-rich permafrost or melting of massive ground ice (Kokelj
51 and Jorgenson, 2013)."

52 ______________________________________________________________________